# CellPLM: Pre-training of Cell Language Model Beyond Single Cells

**Hongzhi Wen**[1][*]**, Wenzhuo Tang**[1][*]**, Xinnan Dai**[1]**, Jiayuan Ding**[1]**, Wei Jin**[2]**,
Yuying Xie**[1]**, Jiliang Tang**[1]
[1]Michigan State University    [2]Emory University
{wenhongz, tangwen2, daixinna, dingjia5, xyy, tangjili}@msu.edu
wei.jin@emory.edu

## Abstract

The current state-of-the-art single-cell pre-trained models are greatly inspired by the success of large language models. They trained transformers by treating genes as tokens and cells as sentences. However, three fundamental differences between single-cell data and natural language data are overlooked: (1) scRNA-seq data are presented as bag-of-genes instead of sequences of RNAs; (2) Cell-cell relations are more intricate and important than inter-sentence relations; and (3) The quantity of single-cell data is considerably inferior to text data, and they are very noisy. In light of these characteristics, we propose a new pre-trained model *CellPLM*, which takes cells as tokens and tissues as sentences. In addition, we leverage spatially-resolved transcriptomic data in pre-training to facilitate learning cell-cell relationships and introduce a Gaussian mixture prior distribution as an additional inductive bias to overcome data limitation. *CellPLM* is the first single-cell pre-trained transformer that encodes cell-cell relations and it consistently outperforms existing pre-trained and non-pre-trained models in diverse downstream tasks, with 100 times higher inference speed on generating cell embeddings than previous pre-trained models.

## 1 Introduction

Next-generation sequencing technologies such as single-cell RNA sequencing (scRNA-seq) have produced vast amounts of data, sparking a surge of interest in developing large-scale pre-trained models for single-cell analysis (Yang et al., 2022; Gong et al., 2023; Shen et al., 2023; Cui et al., 2023; Theodoris et al., 2023). These models seek to capture underlying structures and patterns from unlabeled scRNA-seq data, and can be fine-tuned on specific downstream datasets to deliver accurate predictions and nuanced insights into cellular mechanisms. Particularly, these pre-trained models have been inspired by the success of large language models, such as BERT and GPT (Kenton & Toutanova, 2019; Bubeck et al., 2023), and treat genes as words (tokens) and cells as sentences to train transformers (Vaswani et al., 2017). However, we argue that these approaches may have limitations due to the fundamental differences between single-cell data and natural language data, which have been largely overlooked in existing literature:

*First*, unlike sentences, the scRNA-seq data utilized by existing pre-trained models are not sequential. Before the training stage, RNA sequences have been identified as functional units, i.e., genes. Instead of original sequences, data is denoted as a cell-by-gene count matrix that measures the abundance of individual genes within each cell. This is analogous to the bag-of-words model in natural languages, where the set of genes is fixed, and there is no sequential relationship among them.

*Second*, the relationship between cells is remarkably more intricate and important than that of sentences, since cell-cell communications play an essential role in determining cell states and cell development (Armingol et al., 2021). Additionally, within tissues, there are numerous cells from the same or similar cell lineage, which grants them similar gene expression profiles and hence provides valuable supplementary information for denoising and identifying cell states (Cannoodt et al., 2016; Molho et al., 2022; Street et al., 2018). As a result, many recent methods (Wang et al., 2021; Shao et al., 2022; Xu et al., 2023; Wen et al., 2023) have constructed cell-cell graphs to

---

[*]These authors contribute equally to this work.

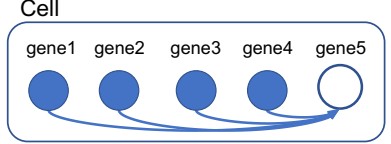

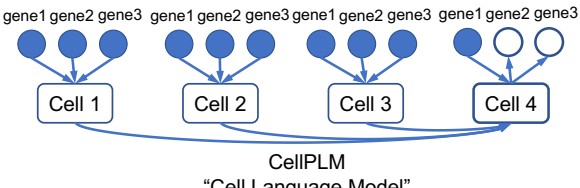

Figure 1: An illustration of the difference in the language models between existing single-cell pre-trained models and *CellPLM*. Existing pre-trained models only consider conditional probability between gene expressions within the same cell, while in *CellPLM*, gene expression distribution is also conditioned on other cells. See details in Section 3.

advance representation learning for single-cell data. Such evidence demonstrates the importance of the cell-cell relationship, which is neglected by existing pre-trained models.

*Third*, the quantity and quality of single-cell datasets are significantly lower than those of natural language data. For comparison, the high-quality filtered English dataset extracted from Common Crawl corpora (Wenzek et al., 2020) consists of 32 billion sentences, whereas the largest collection of single-cell datasets, namely the Human Cell Atlas (Regev et al., 2017), includes less than 50 million cells. To make things worse, single-cell data often suffer from technical artifacts and dropout events (Svensson et al., 2017; Qiu, 2020), as well as significant batch effects between sequencing platforms and experiments (Tran et al., 2020; Argelaguet et al., 2021).

The aforementioned differences introduce distinct challenges that call for new pre-training strategies tailored for single-cell data. To bridge this gap, we propose a novel single-**Cell P**re-trained **L**anguage **M**odel (*CellPLM*), which addresses these challenges from the following perspective: **First**, As shown in Figure 1, *CellPLM* proposes a cell language model to account for cell-cell relations. The cell embeddings are initialized by aggregating gene embeddings since gene expressions are bag-of-word features. **Second**, *CellPLM* leverages a new type of data, spatially-resolved transcriptomic (SRT) data, to gain an additional reference for uncovering cell-cell interactions. Compared to scRNA-seq data, SRT data provide additional positional information for cells. Both types of data are jointly modeled by transformers. **Third**, *CellPLM* introduces inductive bias to overcome the limitation of data quantity and quality by utilizing a Gaussian mixture model as the prior distribution in the latent space. This design can lead to smoother and better cell latent representations (Grønbech et al., 2020; Xu et al., 2023; Jiang et al., 2023). To the best of our knowledge, the proposed *CellPLM* is the first pre-trained transformer framework that encodes inter-cell relations, leverages spatially-resolved transcriptomic data, and adopts a reasonable prior distribution. It is evident from our experiments that *CellPLM* consistently outperforms both pre-trained and non-pre-trained methods across five distinct downstream tasks, with **100 times higher inference speed** on generating cell embeddings compared to existing pre-trained models.

## 2 SINGLE-CELL PRE-TRAINED MODELS

Deep learning methods for single-cell data have garnered significant research interest in recent years (Molho et al., 2022). However, due to the distinct model architectures, the knowledge learned by models is not transferable across tasks. To address this issue, there is an emerging effort (Yang et al., 2022; Gong et al., 2023; Shen et al., 2023; Cui et al., 2023; Theodoris et al., 2023) from the research community to explore the potential of a foundation model that first extracts latent knowledge from unlabeled scRNA-seq data and subsequently generalizes this knowledge to a variety of tasks.

The first such pre-trained model for single-cell data, scBERT (Yang et al., 2022), takes genes as tokens and leverages an efficient transformer (Choromanski et al., 2020) to encode over 16,000 gene tokens for each cell. By randomly masking a fraction of non-zero gene expression values and predicting them based on the remaining data, scBERT effectively learns intricate relationships between genes, leading to improved cellular representation. Later, xTrimoGene (Gong et al., 2023) made two key enhancements to scBERT: pruning zero-expressed genes and improving expression binning strategies by an auto-discretization strategy. These modifications notably enhance scalability and feature resolutions. Another latest preprint, scGPT (Cui et al., 2023), introduces a variant of masked language modeling that mimics the auto-regressive generation in natural language processing, where the masked genes are iteratively predicted according to model's confidence. Unlike the aforementioned models, Geneformer (Theodoris et al., 2023) and tGPT (Shen et al., 2023) completely abandon precise expression levels of genes. Instead, they model the rank of gene expressions and construct sequences of genes according to their relative expression levels within each cell.

The aforementioned models all regard genes as tokens and focus solely on modeling gene relationships within individual cells, neglecting the intercellular information in an organism. In contrast, *CellPLM* overcomes this limitation by introducing a cell language model that extends beyond single cells. Furthermore, by leveraging the spatial information of cells acquired from SRT data, along with a prior Gaussian mixture distribution, the model achieves unparalleled performance on a range of downstream tasks.

## 3 CELL LANGUAGE MODEL BEYOND SINGLE CELLS

In this section, we introduce the concept of the cell language models and detailed implementation of the proposed *CellPLM*. As illustrated in Figure 2, *CellPLM* consists of four modules: a gene expression embedder, an encoder, latent space, and a decoder, which we will demonstrate in Section 3.2. At a higher level, there are two stages in our framework: pre-training and fine-tuning. During pre-training, the model is trained on unlabeled data with a masked language modeling objective. For fine-tuning, the model is first initialized with the pre-trained parameters, and then all of the parameters are fine-tuned using data and labels (if available) from the downstream datasets. We demonstrate the pre-training and fine-tuning framework in Section 3.3 and 3.3, respectively.

### 3.1 CELL LANGUAGE MODEL

Due to the recent achievements of large language models (Bubeck et al., 2023), several studies have drawn inspiration from natural language processing in an attempt to establish a foundation model for single-cell analysis. These studies consider genes as tokens and train transformers on them, aiming to model the conditional probability between gene expressions. Concretely, previous pre-trained models are trained on scRNA-seq data, which are stored in the format of a cell-by-gene matrix $\mathbf{X} \in \mathcal{R}^{N \times k}$, where $N$ is the number of cells, and $k$ is the number of distinct gene types. The value of $\mathbf{X}_{i,j}$ denotes the count of gene $j$ observed in cell $i$, also known as gene expression. The pre-training goal of these models is to estimate a conditional probability distribution, which can be formulated as:

$$p\left(\mathbf{X}_{i,j}|\{\mathbf{X}_{i,o}\}_{o \in \mathcal{O}(i)}\right), j \in \mathcal{U}(i), \tag{1}$$

where $i$ refers to the $i$-th cell and $\mathcal{O}(i)$ is the set of observed genes in cell $i$ whose expressions are known; $\mathcal{U}(i)$ denotes the set of unobserved genes in cell $i$ whose expression will be predicted by the model, typically referring as masked genes. If we consider genes as words, this objective is analogous to the language model in computational linguistics (Bengio et al., 2000), and thus can be named a "gene language model". In this way, the model is trained to capture the intrinsic relations between genes, which can provide prior knowledge for downstream analysis.

However, in Eq. (1), the distribution of unobserved gene expressions only depends on genes within the same cell, while disregarding the information of other cells within the same tissue, which does not align with the inherent nature of biology. Therefore, in *CellPLM*, we provide a different perspective to model scRNA-seq data by treating cells as tokens:

$$p\left(\mathbf{X}_{i,j}|\{\mathbf{X}_{u,v}\}_{(u,v) \in \mathcal{M}^C}\right), (i,j) \in \mathcal{M}, \tag{2}$$

where we denote $\mathcal{M}$ as the set of masked gene expressions in $\mathbf{X}$, and $\mathcal{M}^C$ is the complement, i.e., the set of unmasked expressions. The distribution of a masked entry $\mathbf{X}_{i,j}$ depends on both the observed genes in cell $i$ and genes from other cells that are not masked. We hereby name it as "cell language model", which models the distribution of cellular features beyond single cells. By estimating the conditional probability distribution in Eq. (2), *CellPLM* is trained to capture the intricate relationships that exist between not only genes but also cells.

**From a biology perspective**, there are particularly two types of inter-cell relations that can be beneficial to *CellPLM*. First, within tissues, there are numerous cells from the same or similar cell lineage, which mutually provide valuable supplementary information for denoising and identifying cell states (Cannoodt et al., 2016; Molho et al., 2022; Street et al., 2018). The other type of relations, cell-cell communications, plays an essential role in determining cell development and cell states (Armingol et al., 2021). Existing analysis methods (Hou et al., 2020; Jin et al., 2021; Raredon et al., 2019) have already explored the cell-cell communications on the cell type or cluster levels, while *CellPLM* aims to capture the intricate "language" of cell-cell communications between single cells. Overall, *CellPLM* presents a novel cell language model that aligns well with biological principles and holds great potential to enhance downstream tasks by extracting valuable cellular knowledge from unlabeled single-cell data.

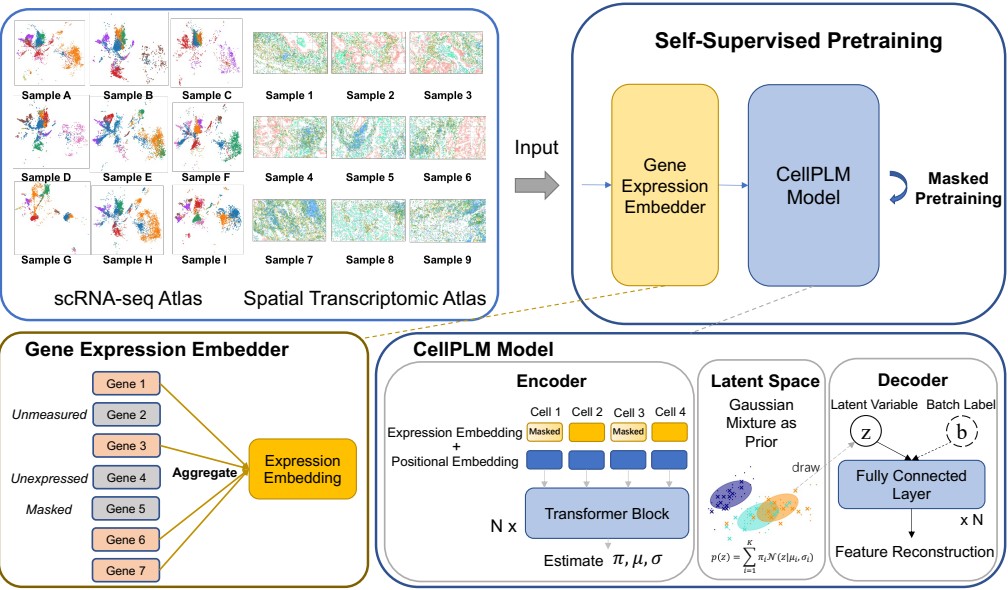

Figure 2: An illustration of the pre-training framework of *CellPLM*. *CellPLM* is pre-trained with cell-level masked language modeling task. The model consists of four modules: a gene expression embedder, a transformer encoder, a Gaussian mixture latent space, and a batch-aware decoder.

## 3.2 MODEL ARCHITECTURE

**Gene Expression Embedder**. The first module in *CellPLM* model is a gene expression embedder, which projects input gene expressions into a low-dimensional cellular feature space. In light of the nature that scRNA-seq is profiled as bag-of-genes features, *CellPLM* learns an embedding vector for each type of gene and then aggregates these gene embeddings according to their expression levels in each cell. Formally speaking, for gene $j \in \{1, ..., k\}$, a randomly-initialized learnable embedding vector $\mathbf{h}_j \in \mathcal{R}^d$ is assigned, where $d$ is the hidden dimension of the encoder layers. The gene expression embedding matrix $\mathbf{E} \in \mathcal{R}^{N \times d}$ is then generated by aggregating gene embeddings according to their expressions:

$$\mathbf{E}_i = \sum_{j=1}^{k} \mathbf{X}_{i,j} \mathbf{h}_j, \tag{3}$$

where $\mathbf{E}_i$ is the $i$-th row vector of $\mathbf{E}$, corresponding to the gene expression embedding for cell $i$. Note that the gene expression matrix $\mathbf{X}$ is a sparse matrix since the zero-rate of scRNA-seq can be up to $90\%$ (Jiang et al., 2022). In addition, unmeasured genes (per sequencing platforms) also lead to zero entries in $\mathbf{X}$. Therefore, when implementing Eq. (3), *CellPLM* leverages sparse operations, which significantly improves memory and computational efficiency. In addition, following the convention (Stuart et al., 2019), we preprocessed $\mathbf{X}$ with library size normalization and log1p transformation before inputting the model.

**Transformer Encoder**. The proposed *CellPLM* follows an encoder-decoder structure, where the encoder is based on transformers (Vaswani et al., 2017). The transformer model was originally developed for processing textual data. It leverages multi-head self-attention mechanisms to capture relationships between input tokens and incorporates positional encoding to represent the token positions. In *CellPLM*, by considering cells as tokens, we can readily apply the transformer model to capture intercellular relationships. When applying the transformer, we consider the embedding at $l$-th layer $\mathbf{H}^{(l)} \in \mathcal{R}^{N \times d}$ as a set of $N$ tokens, where $N$ is the total number of cells in a tissue sample, and $d$ is the hidden dimension. By stacking $L$ transformer layers, *CellPLM* gradually encodes cellular and inter-cellular information into cell embeddings, formulated as:

$$\mathbf{H}^{(l)} = \text{TransformerLayer}^{(l)}(\mathbf{H}^{(l-1)}). \tag{4}$$

In practice, $N$ can scale up to ten thousand, which is out of the capacity of an ordinary transformer. Therefore, we adopt an efficient variant of transformers with linear complexity (i.e., Flowformer (Wu et al., 2022)) for the implementation of transformer layers.

To further inform inter-cellular relations, we incorporate spatial positional information of individual cells from a novel type of data, spatially-resolved transcriptomic (SRT) data. Specifically, SRT data

consists of two parts. One is a gene expression matrix $\mathbf{X} \in \mathcal{R}^{N \times k}$ same as scRNA-seq data, and the other part is a 2D coordinate matrix $\mathbf{C} \in \mathcal{R}^{N \times 2}$. The coordinates denote the center position of each cell within a field-of-view (FOV) where the cells are located (an illustration can be found in Appendix A). This feature helps locate the microenvironment surrounding each cell, providing an additional reference for identifying cell lineage and cell communications, which were introduced in Section 3.1. To encode this extra positional information, we leverage the idea of positional encodings (PE) in transformers. Since sinusoidal PE achieves competitive performance and has lower complexity on SRT data (Wen et al., 2023), we generate a 2D sinusoid PE for cells in SRT data, denoted as $\mathbf{P} \in \mathcal{R}^{N \times d}$, where $\mathbf{P}_i$ is the $d$ dimensional PE vector for cell $i$ (see details in Appendix B). For scRNA-seq data, a randomly initialized $d$-dimensional vector $p'$ is shared among all cells, in order to be unified with SRT data. The initial cell embeddings are now formulated as $\mathbf{H}^{(0)} = \mathbf{E} + \mathbf{P}$, where $\mathbf{E}$ is the expression embeddings from Eq. (3) and $\mathbf{P}$ is the positional embeddings.

**Gaussian Mixture Latent Space**. One of the highlights of *CellPLM* is the design of probabilistic latent space. Prior studies have employed variational autoencoders for single-cell analysis, which typically assumes an isotropic Gaussian distribution as the prior distribution of the latent space (Lopez et al., 2018; Xu et al., 2021). While this approach can effectively remove batch effects, it may also result in a loss of information regarding the underlying biological structure of cell groups. To address this limitation, *CellPLM* incorporates the concept of Gaussian mixture variational encoder (Dilok-thanakul et al., 2016; Yang et al., 2019; Xu et al., 2023), which utilizes a mixture of Gaussians to capture the information of distinct functional groups of cells. Formally, for $i \in \{1, \ldots, N\}$, the generative model of cell $i$ can be formulated as:

$$p(\mathbf{y}_i; \boldsymbol{\pi}) = \text{Multinomial}(\boldsymbol{\pi}),$$

$$p(\mathbf{z}_i \mid \mathbf{y}_i) = \prod_{i=1}^{L} \mathcal{N}\left(\boldsymbol{\mu}_{y_{i,l}}, \text{diag}\left(\boldsymbol{\sigma}_{y_{i,l}}^2\right)\right), \tag{5}$$

$$p_{\theta_{dec}}(\mathbf{x}_i \mid \mathbf{z}_i) = \mathcal{N}\left(\boldsymbol{\mu}_{\mathbf{z}_i}, \sigma^2 \mathbf{I}\right),$$

where $\mathbf{y}_i \in \mathcal{R}^L$ represents the one-hot latent cluster variable and $\boldsymbol{\pi}$ is its prior; $y_{i,l}$ denotes the $l$-th entry of $\mathbf{y}_i$; $\boldsymbol{\mu}_{y_l} \in \mathcal{R}^{d_z}$ and $\boldsymbol{\sigma}_{y_l}^2 \in \mathcal{R}^{d_z \times d_z}$ denote the mean and variance of the $l$-th Gaussian component, respectively; and $\boldsymbol{\mu}_{\mathbf{z}_i} \in \mathcal{R}^k$ and $\sigma^2 \mathbf{I} \in \mathcal{R}^{k \times k}$ denote the posterior mean and variance of expression $\mathbf{x}_i$, respectively. In this work, we assume that $\sigma^2$ is a constant and the posterior mean is parameterized by $\boldsymbol{\mu}_{\mathbf{z}_i} = f_{dec}(\mathbf{z}_i; \theta_{dec})$.

To estimate the posterior of $\mathbf{z}_i$ and $\mathbf{y}_i$, we parameterize the inference process with neural networks, which is detailed in Appendix D. On top of that, a log-evidence lower bound (ELBO) can be derived from this generative model for the optimization purpose (Dilokthanakul et al., 2016). However, as mentioned in Section 3.1, our pre-training framework incorporates a cell language model, where parts of the input gene expression matrix $\mathbf{X}$ are masked. This will result in a modified objective. To formalize the problem, recall that previously we defined the masked set as $\mathcal{M}$. On top of that, we denote $\mathbf{M} \in \mathcal{R}^{N \times k}$ as a mask indicator matrix such that

$$\mathbf{M}_{i,j} = \begin{cases} 1 & \text{if } (i,j) \notin \mathcal{M}, \\ 0 & \text{if } (i,j) \in \mathcal{M}. \end{cases}$$

Let $\tilde{\mathbf{X}} \in \mathcal{R}^{N \times k}$ be the masked gene expression matrix given by the element-wise multiplication $\tilde{\mathbf{X}} = \mathbf{M} \odot \mathbf{X}$. The objective of cell language model with Gaussian mixture prior, i.e., a denoising variational lower bound (Im Im et al., 2017), can be formulated as:

$$\mathcal{L}_{\text{CellLM}} = \mathbb{E}_{q(\mathbf{Z}, \mathbf{Y} \mid \tilde{\mathbf{X}})} \mathbb{E}_{p(\tilde{\mathbf{X}} \mid \mathbf{X})} \left[ \ln \frac{p_\theta(\mathbf{X}, \mathbf{Z}, \mathbf{Y})}{q_\eta(\mathbf{Z}, \mathbf{Y} \mid \tilde{\mathbf{X}})} \right] \tag{6}$$

$$= \underbrace{\mathbb{E}_{q_{\eta_{enc}}(\mathbf{Z} \mid \tilde{\mathbf{X}})} \mathbb{E}_{p(\tilde{\mathbf{X}} \mid \mathbf{X})} [\log p_{\theta_{dec}}(\mathbf{X} \mid \mathbf{Z})]}_{\mathcal{L}_{\text{recon}}} - \underbrace{\mathbb{E}_{q_{\eta_\pi}(\mathbf{Y} \mid \mathbf{Z})} \left[ \text{KL}\left(q_{\eta_{enc}}(\mathbf{Z} \mid \tilde{\mathbf{X}}) \| p(\mathbf{Z} \mid \mathbf{Y})\right) \right]}_{\mathcal{L}_{\text{cond}}}$$

$$- \underbrace{\mathbb{E}_{q_{\eta_{enc}}(\mathbf{Z} \mid \tilde{\mathbf{X}})} [\text{KL}\left(q_{\eta_\pi}(\mathbf{Y} \mid \mathbf{Z}) \| p(\mathbf{Y})\right)]}_{\mathcal{L}_{\text{Y}}}.$$

Similar to previous works (Dilokthanakul et al., 2016), we refer to the three terms in Eq. (6) as reconstruction term $\mathcal{L}_{\text{recon}}$, conditional prior term $\mathcal{L}_{\text{cond}}$ and $\mathbf{Y}$ prior term $\mathcal{L}_{\text{Y}}$. The approximation and estimation of the denoising variational lower bound are specified in Section 3.3.

**Batch-aware Decoder**. The decoder in *CellPLM* operates by decoding each cell individually, given that the tissue context has already been encoded into the latent space by the encoder. The decoder's purpose is twofold: to reconstruct masked features and to help remove batch effects from the latent space. In order to accomplish this goal, the decoder stacks several feed-forward layers (FFLayers) atop the input of latent variables $\mathbf{z}$, and a batch embedding, denoted as $\mathbf{b} \in \mathcal{R}^{d_z}$. Specifically, for each cell, the batch embedding is loaded from a learnable lookup table as $\mathbf{b} = \mathrm{LookUp}(b)$, where $b$ is the label indicating the specific tissue sample (or FOV for SRT data) from which the cell has been drawn. By feeding the batch label to the decoder, a batch-effect-free latent space can be achieved, as empirically evidenced in scVI (Lopez et al., 2018). The decoder can thus be formulated as:

$$\mathbf{h}^{(0)} = \mathbf{z} + \mathbf{b}, \quad \mathbf{h}^{(l)} = \mathrm{FFLayer}^{(l)}(\mathbf{h}^{(l-1)}),$$

where $l$ indicates the number of the layer, $\mathbf{h}^{(l)}$ is the hidden vector of layer $l \in (1..L-1)$, and $L$ is the total number of fully connected layers. The dimension of the last layer is different from the previous layers because the last layer is considered as an output layer, with $\mathbf{h}^L \in \mathcal{R}^k$, where $k$ is the size of gene sets in the gene expression matrix $\mathbf{X} \in \mathcal{R}^{N \times k}$.

### 3.3 MODEL PRE-TRAINING & FINE-TUNING

**Pre-training.** The pre-training of *CellPLM* follows a cell language modeling objective, as demonstrated in Eq. (6). Specifically, given a batch of cell tokens as input, we first decide which cells should be masked. Instead of completely masking these cell tokens, we selectively mask a certain percentage of the gene expressions within them. This allows the model to recover underlying correlations between cells, as proposed in a recent preprint, SpaFormer (Wen et al., 2023). A significant concern in *CellPLM* is the disparity in the number of genes measured by different sequencing platforms. Notably, the gap between scRNA-seq and SRT can be substantial, ranging from 1,000 to 30,000. Taking this into consideration, ***CellPLM* only masks the expression of genes that are measured in each dataset**, implying that the reconstruction loss is calculated exclusively on these measured genes. When optimizing the denoising variational lower bound in Eq. (6), we apply reparameterization trick and Monte Calo sampling, as proposed in VAE (Kingma & Welling, 2014). Furthermore, under the independent Gaussian assumption, we reformulate and estimate the reconstruction term $\mathcal{L}_{\mathrm{recon}}$ in Eq. (6) with a mean squared error (MSE). Therefore, the pre-training loss function of *CellPLM* can be formulated as:

$$\mathcal{L}_{\mathrm{MSE}} = \left\| \mathbf{M} \odot \left( \mathbf{H}^{(L)} - (1 - \mathbf{M}) \odot \mathbf{X} \right) \right\|_F^2, \mathcal{L}_{\mathrm{pretrain}} = \mathcal{L}_{\mathrm{MSE}} + \mathcal{L}_{\mathrm{cond}} + \mathcal{L}_{\mathrm{Y}}, \quad (7)$$

where $\odot$ signifies element-wise multiplication, $\mathbf{H}^{(L)} \in \mathcal{R}^{N \times k}$ is the output from the decoder, $\mathbf{X}$ and $\mathbf{M}$ are the ground-truth gene expression matrix and the mask indicator matrix respectively, as defined above. $\mathcal{L}_{\mathrm{cond}}$ and $\mathcal{L}_{\mathrm{Y}}$ are derived from Eq. (6).

**Task-specific Fine-tuning**. When fine-tuning *CellPLM*, the model is first initialized with the pretrained parameters. In downstream tasks that require gene expressions as output, the pre-trained decoder can be directly leveraged, and the batch embedding is set to the mixture of all pre-training batches. Otherwise, the decoder will be replaced with a task-specific head. The entire model is then fine-tuned with task-specific loss functions, which helps align the general knowledge of the model to the specific downstream task. For example, in the spatial transcriptomic imputation task, the entire pre-trained model can do zero-shot inference. It can also be fine-tuned on a query SRT dataset and a reference scRNA-seq dataset, where two datasets are sampled from the same type of tissue. In this case, the loss function remains the same as Eq.(7). After fine-tuning on these datasets, *CellPLM* fits the data distribution of the target tissue and can readily perform imputation. The design and implementation of heads and loss functions for other downstream tasks are elucidated in Appendix F.

## 4 EXPERIMENT

*CellPLM* is first pre-trained on more than 9 Million scRNA-seq cells and 2 Million SRT cells, with the masked language modeling objective, demonstrated in Section 3.3. The model consists of over 80 million parameters and the pre-training was finished in less than 24 hours on a GPU server with 8 Nvidia Tesla v100 16GB cards. The hyperparameters, datasets, and reproducibility information for pre-trained models are detailed in Appendix E.

In the following sections, we evaluate the performance of *CellPLM* on various downstream tasks, including zero-shot clustering, scRNA-seq denoising, spatial transcriptomic imputation, cell type annotation, and perturbation prediction. With the selected tasks, we aim to answer the following research questions:

**RQ1:** Is *CellPLM* capable of transferring pre-train knowledge to a brand new dataset?

**RQ2:** Does *CellPLM* provide better cell representations than other pre-trained and non-pre-trained models?

**RQ3:** Does *CellPLM* succeed in jointly modeling scRNA-seq and SRT data?

## 4.1 PRELIMINARY STUDY: ZERO-SHOT CLUSTERING

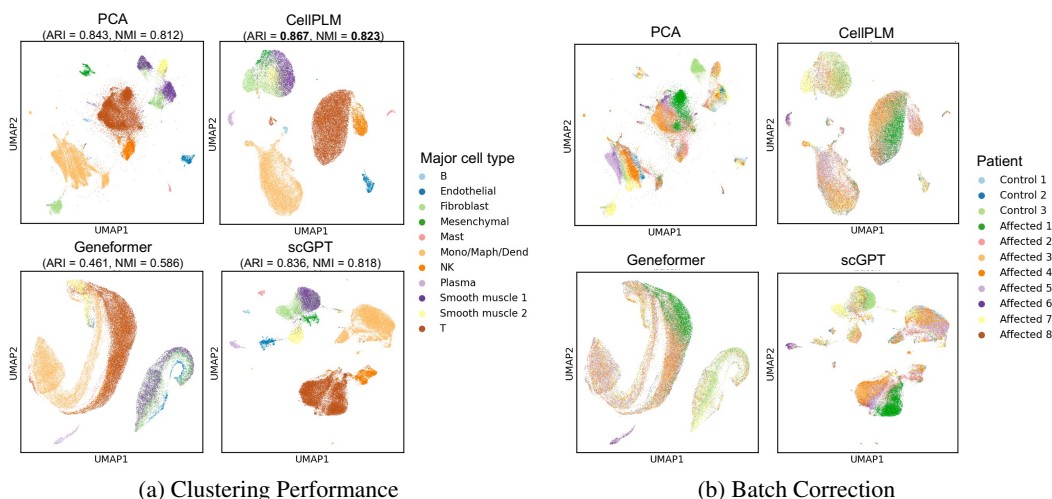

(a) Clustering Performance        (b) Batch Correction

Figure 3: *CellPLM* readily removes patient batch effect and provides accurate cell clustering results without fine-tuning.

| Geneformer | scGPT | *CellPLM* |
|:---:|:---:|:---:|
| 428.24 | 129.19 | **0.85** |

Table 1: Inference time(s) for querying $48,082$ cells on an A100 40GB GPU. Preprocessing functions and decoders are not included in this test. Due to GPU memory capacity, the batch size of Geneformer and scGPT is set to 256 and 64 respectively, while the batch size of *CellPLM* is $48,082$.

To evaluate the transferability of the pre-trained model, we extract cell embeddings from the pre-trained encoder on a public dataset from Li et al. (2020), which is not included in pre-train data. In addition to *CellPLM*, we include three baselines, i.e., PCA, Geneformer (Theodoris et al., 2023) and scGPT(Cui et al., 2023). PCA refers to the first 512 PCs of log-normalized expressions from $4500$ highly variable genes (the number of PCs equals the embedding size of scGPT and *CellPLM*). This is a common embedding method on scRNA-seq data. Geneformer and scGPT are two recently published pre-trained models that are capable of generating cell embeddings. Figure 3a illustrates how well the embeddings are aligned with curated cell type labels, and Figure 3b demonstrates models' ability to remove technical artifacts and mix biological signals from different experiments. Notably, the clustering result of *CellPLM*'s embedding achieves the highest ARI and NMI with respect to the ground-truth cell type labels. From the visualization, it is also clear that *CellPLM* possesses smoother latent space than others, which is attributed to our Gaussian mixture prior distribution for pre-training. We also notice that *CellPLM* is over $100$ times faster than other pre-trained models that conduct self-attention among gene tokens, as shown in Table 1. Overall, this preliminary study addresses **RQ1** and **RQ2**, and indicates that *CellPLM* can readily transfer pre-trained knowledge to new datasets in removing batch effect and generating high-quality cell embeddings, at extraordinarily high inference speed.

## 4.2 TASK 1: SCRNA-SEQ DENOISING

Given that single-cell RNA-Seq protocols capture only a subset of the mRNA molecules within individual cells, the resulting measurements exhibit substantial technical noise (Grün et al., 2014). Therefore, we consider denoising power as the most desired and essential property for a single-cell foundation model. The goal of the denoising task is to estimate the true expression level of each gene in each cell from a noisy observation. To assess the denoising efficacy of *CellPLM*, we conduct an evaluation on two single-cell RNA-Seq datasets, i.e., PBMC 5K and Jurkat from 10x Genomics (lin, a). These two datasets were excluded from pre-training. Following the setting of scGNN (Wang et al., 2021), we apply a random flipping process to a subset of non-zero entries, transforming them into zeros to simulate the dropout effects. We compare *CellPLM* against a broad range of contemporary approaches, including DeepImpute (Arisdakessian et al., 2019), scGNN2.0 (Gu et al., 2022), SAVER (Huang et al., 2018), DCA (Eraslan et al., 2019), scVI (Lopez et al., 2018), MAGIC (Van Dijk et al., 2018), scImpute (Li & Li, 2018) and scGPT (Cui et al., 2023). We evaluate scRNA-seq denoising performance based on two popular regression metrics, i.e., Root Mean Square Error (RMSE) and Mean Absolute Error (MAE), to measure the degree of similarity between predicted gene expression and the actual ones. More details pertaining to these methods, the fine-tuning of *CellPLM*, and the evaluation metrics can be found in Appendix F.1.

It is evident that the fine-tuned *CellPLM* consistently exhibits superior performance compared to all baseline models on both datasets. Note that even under the **zero-shot setting**, *CellPLM* shows satisfactory results that surpass the majority of baselines (6 out of 8) on each dataset. These observations answer the question of **RQ1** and **RQ2**. As a powerful denoising model, *CellPLM* can serve as a foundation for other downstream analyses.

| Model | PBMC 5K | | Jurkat | |
| --- | --- | --- | --- | --- |
| | RMSE ($\downarrow$) | MAE ($\downarrow$) | RMSE ($\downarrow$) | MAE ($\downarrow$) |
| DeepImpute | $1.168 \pm 0.018$ | $1.051 \pm 0.025$ | $0.786 \pm 0.006$ | $0.557 \pm 0.003$ |
| scGNN 2.0 | $1.376 \pm 0.015$ | $1.237 \pm 0.019$ | $1.001 \pm 0.016$ | $0.917 \pm 0.021$ |
| GraphSCI | $1.068 \pm 0.007$ | $0.924 \pm 0.009$ | $0.659 \pm 0.030$ | $0.481 \pm 0.024$ |
| SAVER | $0.884 \pm 0.001$ | $0.748 \pm 0.001$ | $0.569 \pm 0.001$ | $0.472 \pm 0.001$ |
| DCA | $0.775 \pm 0.002$ | $0.621 \pm 0.002$ | $0.423 \pm 0.001$ | $0.351 \pm 0.001$ |
| scVI | $0.777 \pm 0.005$ | $0.623 \pm 0.004$ | $0.416 \pm 0.001$ | $0.344 \pm 0.002$ |
| MAGIC | $0.793 \pm 0.001$ | $0.639 \pm 0.001$ | $0.424 \pm 0.001$ | $0.351 \pm 0.002$ |
| scImpute | $1.170 \pm 0.003$ | $1.002 \pm 0.001$ | $0.624 \pm 0.002$ | $0.529 \pm 0.001$ |
| scGPT (fine-tuned) | $0.901 \pm 0.001$ | $0.565 \pm 0.001$ | $0.711 \pm 0.001$ | $0.498 \pm 0.001$ |
| *CellPLM* (zero-shot) | $0.854 \pm 0.001$ | $0.692 \pm 0.000$ | $0.517 \pm 0.001$ | $0.426 \pm 0.000$ |
| *CellPLM* (from scratch) | $0.761 \pm 0.009$ | $0.571 \pm 0.011$ | $0.395 \pm 0.003$ | $0.320 \pm 0.003$ |
| *CellPLM* (fine-tuned) | $\mathbf{0.725 \pm 0.001}$ | $\mathbf{0.551 \pm 0.001}$ | $\mathbf{0.391 \pm 0.001}$ | $\mathbf{0.320 \pm 0.001}$ |

Table 2: (*Task 1*) The scRNA-seq denoising performance on the PBMC 5K and Jurkat datasets.

## 4.3 TASK 2: SPATIAL TRANSCRIPTOMIC IMPUTATION

Spatially resolved transcriptomics has revolutionized single-cell analysis by incorporating physical locations along with gene expression, leading to exciting breakthroughs. However, as a tradeoff for the highly detailed spatial resolution, spatial transcriptomic data at the cellular level typically cover less than $1,000$ genes, which poses challenges in data analysis. To assess the potential benefits of the pre-trained model in the given task, we evaluate *CellPLM* on two spatial transcriptomic datasets at single-cell resolution, i.e., Lung2 and Liver2 from lin (b) (the whole study is not included in our pre-train data). Following the setting of baselines including SpaGE (Abdelaal et al., 2020), stPlus (Shengquan et al., 2021), gimVI (Lopez et al., 2019) and Tangram (Biancalani et al., 2021), we impute the unmeasured genes of the SRT dataset utilizing a scRNA-seq dataset as reference. We identify the testing gene set in SRT data by stratified sampling according to gene sparsity (Avşar & Pir, 2023) and hold out those genes in the fine-tuning stage. To evaluate the accuracy of spatial transcriptomic imputation, we employ the Pearson correlation coefficient (Corr) and cosine similarity (Cosine) to measure the degree of similarity between the predicted spatial gene expressions and the corresponding ground-truth expression values. Details of the implementation and the evaluation metrics are presented in Appendix F.2.

Remarkably, the fine-tuned *CellPLM* takes the lead on both datasets, effectively addressing the research question **RQ1** and **RQ3**. However, when training from scratch on these datasets, *CellPLM* hardly converges. This indicates the pre-training information is necessary for *CellPLM* to impute the SRT data.

| | Lung2 | | Liver2 | |
|---|---|---|---|---|
| Model | Corr ($\uparrow$) | Cosine ($\uparrow$) | Corr ($\uparrow$) | Cosine ($\uparrow$) |
| SpaGE | $0.227 \pm 0.011$ | $0.352 \pm 0.015$ | $0.253 \pm 0.014$ | $0.376 \pm 0.005$ |
| stPlus | $0.177 \pm 0.021$ | $0.360 \pm 0.014$ | $0.224 \pm 0.010$ | $0.399 \pm 0.012$ |
| gimVI | $0.130 \pm 0.010$ | $0.325 \pm 0.010$ | $0.163 \pm 0.019$ | $0.338 \pm 0.010$ |
| Tangram | $0.123 \pm 0.005$ | $0.285 \pm 0.008$ | $0.168 \pm 0.024$ | $0.309 \pm 0.008$ |
| *CellPLM* (zero-shot) | $0.119 \pm 0.024$ | $0.327 \pm 0.011$ | $0.141 \pm 0.013$ | $0.322 \pm 0.145$ |
| *CellPLM* (from scratch) | $0.058 \pm 0.020$ | $0.370 \pm 0.013$ | $0.024 \pm 0.039$ | $0.352 \pm 0.011$ |
| *CellPLM* (fine-tuned) | $\mathbf{0.318 \pm 0.015}$ | $\mathbf{0.481 \pm 0.011}$ | $\mathbf{0.328 \pm 0.011}$ | $\mathbf{0.481 \pm 0.010}$ |

Table 3: (*Task 2*) The results of spatial tanscriptomic imputation on the Lung2 and Liver2 datasets.

### 4.4 TASK 3: CELL TYPE ANNOTATION

Cell type annotation is another important task in single-cell analysis as it enables the identification and characterization of distinct cell populations within a tissue or organism. The objective of this task is to classify the type of cells from query datasets according to the annotations in reference datasets. Here we follow the suggestion of Cui et al. (2023) to include hPancreas (Chen et al., 2023) and Multiple Sclerosis (MS) (Schirmer et al., 2019) datasets. More details about the datasets, baseline methods, the fine-tuning of *CellPLM* can be found in Appendix G.

| | MS | | hPancreas | |
|---|---|---|---|---|
| | F1 ($\uparrow$) | Precision ($\uparrow$) | F1 ($\uparrow$) | Precision ($\uparrow$) |
| CellTypist | $0.667 \pm 0.002$ | $0.693 \pm 0.001$ | $0.708 \pm 0.023$ | $0.736 \pm 0.025,$ |
| ACTINN | $0.628 \pm 0.012$ | $0.634 \pm 0.009$ | $0.705 \pm 0.005$ | $0.709 \pm 0.006$ |
| SingleCellNet | $0.637 \pm 0.001$ | $0.700 \pm 0.001$ | $0.739 \pm 0.006$ | $\mathbf{0.761 \pm 0.004}$ |
| TOSICA* | $0.578$ | $0.664$ | $0.656$ | $0.661$ |
| scBERT* (fine-tuned) | $0.599$ | $0.604$ | $0.685$ | $0.699$ |
| scGPT* (fine-tuned) | $0.703$ | $0.729$ | $0.718$ | $0.735$ |
| *CellPLM* (from scratch) | $0.709 \pm 0.007$ | $0.732 \pm 0.015$ | $0.689 \pm 0.034$ | $0.682 \pm 0.037$ |
| *CellPLM* (fine-tuned) | $\mathbf{0.766 \pm 0.007}$ | $\mathbf{0.803 \pm 0.008}$ | $\mathbf{0.749 \pm 0.010}$ | $0.753 \pm 0.010$ |

Table 4: (*Task 3*) The results of cell type annotation on MS and hPancreas dataset. "*" indicates results directly taken from Cui et al. (2023).

The empirical results presented in Table 4 indicate that *CellPLM* learns a well-represented and generalizable cellular embedding, achieving considerably large improvement on the cell type annotation task. This again confirms our positive answer to the research questions **RQ1** and **RQ2**.

In addition to these major results, we also conduct analysis and experiments on the gene level. The results show that *CellPLM* learns meaningful gene representations and can benefit genetic perturbation prediction. Due to space limits, we leave these discussions in Appendix H.2 and F.3. Finally, to verify the effectiveness of our proposed transformer encoder and the mixture of Gaussian latent distribution, we conduct a series of ablation studies, presented in Appendix I. Through the ablation studies, we confirm that our *CellPLM* model can capture the relationships between cells via the transformer encoder and enhance the performance of downstream tasks, generating more robust and useful cell representations through appropriate prior distributions.

## 5 CONCLUSION

In this work, we propose cell language model, a novel paradigm of single-cell pre-trained model, which aligns well with the fundamental characteristics of single-cell data. This has led to *CellPLM*, the first pre-trained transformer framework that encodes inter-cell relations, leverages spatially-resolved transcriptomic data and adopts a reasonable prior distribution. Our experiments on various downstream tasks demonstrate the power of *CellPLM*, which has great potential to facilitate future research in single-cell biology.

**Reproducibility Statement**: All the data we used in this study are publicly available data. The data sources are specified in the appendix. The checkpoint of our pre-trained is released on our Github[1] repository, as well as the source codes for fine-tuning and zero-shot experiments. All the hyperparameters are specified either in the script files or in the appendix.

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

# Appendix for CellPLM: Pre-training of Cell Language Model Beyond Single Cells

## A SPATIALLY-RESOLVED TRANSCRIPTOMIC DATA

Recently, spatial transcriptomic technologies are developed to spatially resolve transcriptomics profiles Ståhl et al. (2016); Merritt et al. (2020). With spatial transcriptomics data, researchers can learn the spatial context of cells and cell clusters within a tissue Burgess (2019). The major technologies/platforms for spatial transcriptomics are Visium by 10x Ståhl et al. (2016), GeoMx Digital Spatial Profiler (DSP) Merritt et al. (2020) by NanoString and CosMx Spatial Molecular Imager (SMI) by NanoString, MERFISH, Vizgen, Resolve, Rebus, and molecular cartography. 10x Visium does not profile at single-cell resolution, and while GeoMx DSP is capable of single-cell resolution through user-drawn profiling regions, the scalability is limited. The most recent platform, CosMx Spatial Molecular Imager (SMI) He et al. (2022), can profile consistently at single-cell and even sub-cellular resolution. CosMx SMI follows much of the initial protocol as GeoMx DSP, with barcoding and ISH hybridization. However, the SMI instrument performs 16 cycles of automated cyclic readout, and in each cycle, the set of barcodes (readouts) are UV-cleaved and removed. These cycles of hybridization and imaging yield spatially resolved profiling of RNA and protein at single-cell ($\sim 10\mu m$) and subcellular ($\sim 1\mu m$) resolution. In this work, we use two published and one unpublished dataset produced by the CosMx platform. In order to obtain the cellular level gene expression, CellPose Stringer et al. (2021) software is applied to conduct cell segmentation.

To give a concrete example, we provide a sample field-of-view (FOV) in Fig. 4. Pre-selected types of RNA molecules are captured by the molecular imager, which are denoted as white dots in the figures. Colors in the first sub-figure indicate the protein molecules that are stained. These proteins contribute to the cell segmentation process, which results in the second sub-figure. The final output from the pipeline consists of the position of each cell and a cell-by-gene count matrix, which is produced by counting the number of RNA molecules within each cell. The difference between scRNA-seq and SRT data is further demonstrated in Fig. 5.

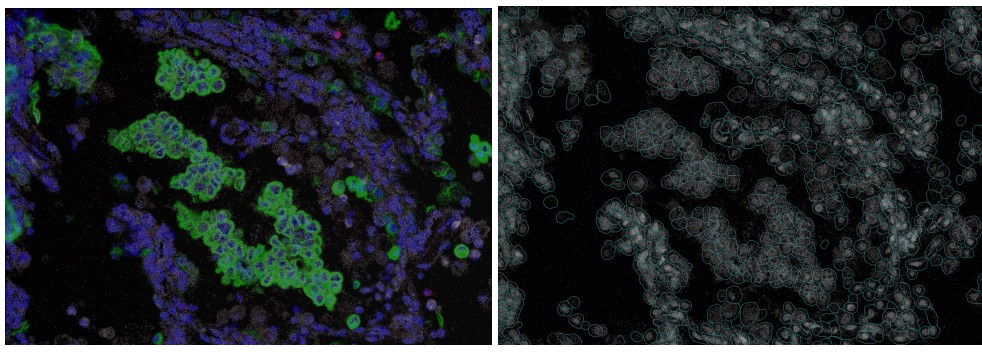

(a) Visualization of molecular image.  (b) Visualization of cell segmentation.

Figure 4: (a) A sample image of protein and RNA molecules. (b) A sample image of segmented cells.

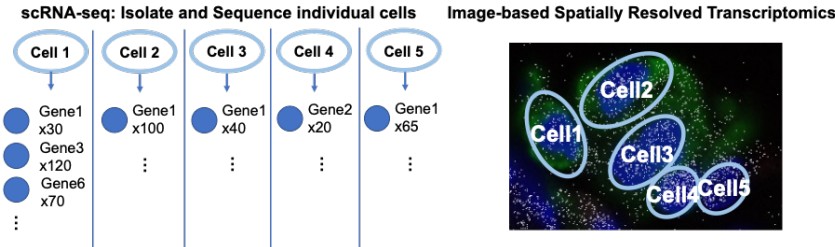

Figure 5: An illustration of the difference between scRNA-seq and SRT data.

## B    2D SINUSOID POSITIONAL ENCODINGS

Since 2D sinusoidal PE achieves a competitive performance and has a lower complexity on SRT data Wen et al. (2023), in our transformer encoer, we generate a sinusoidal PE for cells in SRT data, formulated as:

$$
\begin{aligned}
\mathrm{PE}_{(x,y,2i)} &= \sin\left(x/10000^{4i/d}\right), \mathrm{PE}_{(x,y,2i+1)} = \cos\left(x/10000^{4i/d}\right), \\
\mathrm{PE}_{(x,y,2j+d/2)} &= \sin\left(y/10000^{4j/d}\right), \mathrm{PE}_{(x,y,2j+1+d/2)} = \cos\left(y/10000^{4j/d}\right),
\end{aligned}
\tag{8}
$$

where $d$ is the total dimension of positional encoding, $i,j \in [0, d/4)$ specify a specific feature dimension. Let $\tilde{\mathbf{C}} \in \mathcal{R}^{N \times 2}$ be a normalized coordinate matrix, where we normalize and truncate coordinates in $\mathbf{C}$ to integers ranging in $[0, 100)$. $x, y$ then refer to the spatial coordinates from $\tilde{\mathbf{C}}$, e.g., $x = \tilde{\mathbf{C}}_{t,0}$ and $y = \tilde{\mathbf{C}}_{t,1}$ for cell $t$. In this way, we generate a PE matrix $\mathbf{P} \in \mathcal{R}^{N \times d}$ for every cell in SRT data, where $\mathbf{P}_i$ is the PE vector for cell $i$. Meanwhile, for scRNA-seq data, a randomly initialized $d$-dimensional vector $p'$ is shared among all cells, which also results in a placeholder PE matrix $\mathbf{P}$.

## C    BROADER IMPACT

Our method lies in an emerging and important application area, single-cell analysis. Especially, we leverage a novel type of single-cell data, Spatially Resolved Transcriptomics (SRT). SRT is a rapidly developing technology that allows scientists to map the gene expression of individual cells in their tissue environment. It combines traditional imaging techniques with transcriptome analysis to provide a spatially resolved, high-resolution view of gene expression in complex tissues. Essentially, single-cell technologies and SRT allow researchers to see where specific genes are being expressed within a tissue sample, which can help them better understand cellular interactions and the function of specific genes in complex biological systems.

We evaluate our method on various downstream tasks and the empirical results demonstrate the practical value of our method. Specifically, scRNA-seq Denoising improves the data quality of scRNA-seq data, which often suffer from technical artifacts and dropout events Svensson et al. (2017); Qiu (2020), as well as significant batch effects between sequencing platforms and experiments Tran et al. (2020); Argelaguet et al. (2021). SRT imputation helps to obtain more precise cell state profiles for SRT data, while also resulting in more accurate integration and clustering between SRT data and scRNA-seq data. Perturbation prediction has great clinical value to aid in drug design and disease mechanism research.

While our work offers a significant contribution to the field of single-cell analysis, there are potential negative societal impacts that are important to consider: one of the primary potential negative societal impacts is privacy and data security. Single-cell analysis involves working with sensitive genetic information which, if mishandled, could lead to breaches in privacy and the misuse of personal data. Another potential negative impact is over-reliance on automated analysis. The complexity of single-cell data requires careful interpretation, and the risk of false-positive or false-negative results may be elevated due to computational errors or algorithmic biases. It is crucial to remember that these tools should serve as aids to human understanding and decision-making rather than replacements.

As single-cell technologies continue to evolve, it is critical that we continue to consider and address these broader societal impacts. Moving forward, it is crucial that our work is coupled with ongoing discussions on best practices in data management, privacy protection, and equitable access to technology. This includes strengthening collaborations with ethicists, policymakers, and regulatory bodies to navigate these complex issues.

## D    DENOISING VARIATIONAL LOWER BOUND FOR MASKED LANGUAGE MODELING

One of the highlights of *CellPLM* is the design of probabilistic latent space. Prior studies have employed variational autoencoders for single-cell analysis, which typically assumes an isotropic Gaussian distribution as the prior distribution of the latent space (Lopez et al., 2018; Xu et al., 2021).

While this approach can effectively remove batch effects, it may also result in a loss of information regarding the underlying biological structure of cell groups. To address this limitation, *CellPLM* incorporates the concept of Gaussian mixture variational encoder (Dilokthanakul et al., 2016; Yang et al., 2019; Xu et al., 2023), which utilizes a mixture of Gaussians to capture the information of distinct functional groups of cells. Formally, for $i \in \{1, \ldots, N\}$, the generative model of cell $i$ can be formulated as:

$$p(\mathbf{y}_i; \boldsymbol{\pi}) = \text{Multinomial}(\boldsymbol{\pi}),$$

$$p(\mathbf{z}_i \mid \mathbf{y}_i) = \prod_{i=1}^{L} \mathcal{N}\left(\boldsymbol{\mu}_{y_{i,l}}, \text{diag}\left(\boldsymbol{\sigma}^2_{y_{i,l}}\right)\right), \tag{9}$$

$$p_{\theta_{dec}}(\mathbf{x}_i \mid \mathbf{z}_i) = \mathcal{N}\left(\boldsymbol{\mu}_{\mathbf{z}_i}, \sigma^2\mathbf{I}\right),$$

where $\mathbf{y}_i \in \mathcal{R}^L$ represents the one-hot latent cluster variable and $\boldsymbol{\pi}$ is its prior; $y_{i,l}$ denotes the $l$-th entry of $\mathbf{y}_i$; $\boldsymbol{\mu}_{yl} \in \mathcal{R}^{d_z}$ and $\boldsymbol{\sigma}^2_{yl} \in \mathcal{R}^{d_z \times d_z}$ denote the mean and variance of the $l$-th Gaussian component, respectively; and $\boldsymbol{\mu}_{z_i} \in \mathcal{R}^k$ and $\sigma^2\mathbf{I} \in \mathcal{R}^{k \times k}$ denote the posterior mean and variance of expression $\mathbf{x}_i$, respectively. In this work, we assume that $\sigma^2$ is a constant and the posterior mean is parameterized by $\boldsymbol{\mu}_{z_i} = f_{dec}(\mathbf{z}_i; \theta_{dec})$.

To estimate the posterior of $\mathbf{z}_i$ and $\mathbf{y}_i$, we parameterize the inference process with neural networks. Specifically, we assume that the cluster variables $\mathbf{y}$ are independent of the expression $\mathbf{x}_i$ condition on latent variables $\mathbf{z}_i$. The inference model can be formulated as:

$$q_{\eta_\mu, \eta_\sigma}(\mathbf{z}_i \mid \mathbf{x}_i) = \mathcal{N}\left(\hat{\boldsymbol{\mu}}_i, \text{diag}\left(\hat{\boldsymbol{\sigma}}^2_i\right)\right),$$
$$q_{\eta_\pi}(\mathbf{y}_i \mid \mathbf{z}_i) = \text{Multinomial}(\hat{\boldsymbol{\pi}}_i), \tag{10}$$

where the estimations are given by

$$\mathbf{h}_i = f_{enc}(\mathbf{x}_i; \eta_{enc}),$$
$$\hat{\boldsymbol{\mu}}_i = f_\mu(\mathbf{h}_i; \eta_\mu),$$
$$\log\left(\hat{\boldsymbol{\sigma}}^2_i\right) = f_\sigma(\mathbf{h}_i; \eta_\sigma), \tag{11}$$
$$\hat{\boldsymbol{\pi}}_i = f_\pi(\mathbf{z}_i; \eta_\pi).$$

Here $f_{enc}(\cdot; \eta_{enc})$ represents the transformer encoder, $f_\mu(\cdot; \eta_\mu)$, $f_\sigma(\cdot; \eta_\sigma)$ and $f_\pi(\cdot; \eta_\pi)$ are neural networks. A log-evidence lower bound (ELBO) can be derived from this generative model for the optimization purpose (Dilokthanakul et al., 2016). However, as mentioned in Section 3.1, our pre-training framework incorporates a cell language model, where parts of the input gene expression matrix $\mathbf{X}$ are masked. This will result in a modified objective. To formalize the problem, recall that previously we defined the masked set as $\mathcal{M}$. On top of that, we denote $\mathbf{M} \in \mathcal{R}^{N \times k}$ as a mask indicator matrix such that

$$\mathbf{M}_{i,j} = \begin{cases} 1 & \text{if } (i,j) \notin \mathcal{M}, \\ 0 & \text{if } (i,j) \in \mathcal{M}. \end{cases}$$

Let $\tilde{\mathbf{X}} \in \mathcal{R}^{N \times k}$ be the masked gene expression matrix given by the element-wise multiplication $\tilde{\mathbf{X}} = \mathbf{M} \odot \mathbf{X}$. The objective of cell language model with Gaussian mixture prior, i.e., a denoising variational lower bound (Im Im et al., 2017), can be formulated as:

$$\mathcal{L}_{\text{CellLM}} = \mathbb{E}_{q(\mathbf{Z}, \mathbf{Y}|\tilde{\mathbf{X}})} \mathbb{E}_{p(\tilde{\mathbf{X}}|\mathbf{X})} \left[ \ln \frac{p_\theta(\mathbf{X}, \mathbf{Z}, \mathbf{Y})}{q_\eta(\mathbf{Z}, \mathbf{Y} \mid \tilde{\mathbf{X}})} \right] \tag{12}$$

$$= \underbrace{\mathbb{E}_{q_{\eta_{enc}}(\mathbf{Z}|\tilde{\mathbf{X}})} \mathbb{E}_{p(\tilde{\mathbf{X}}|\mathbf{X})} [\log p_{\theta_{dec}}(\mathbf{X} \mid \mathbf{Z})]}_{\mathcal{L}_{\text{recon}}} - \underbrace{\mathbb{E}_{q_{\eta_\pi}(\mathbf{Y}|\mathbf{Z})} \left[ \text{KL}\left( q_{\eta_{enc}}(\mathbf{Z} \mid \tilde{\mathbf{X}}) \| p(\mathbf{Z} \mid \mathbf{Y}) \right) \right]}_{\mathcal{L}_{\text{cond}}}$$

$$- \underbrace{\mathbb{E}_{q_{\eta_{enc}}(\mathbf{Z}|\tilde{\mathbf{X}})} [\text{KL}(q_{\eta_\pi}(\mathbf{Y} \mid \mathbf{Z}) \| p(\mathbf{Y}))]}_{\mathcal{L}_{\text{Y}}}.$$

# E  PRE-TRAINING SETTINGS

## E.1  HYPERPARAMETER SETTINGS

We pre-trained *CellPLM* model with the hyperparameters specified in Table 5.

|  | CellPLM |
|---|---|
| encoder hidden dim | 1024 |
| encoder layers | 4 |
| latent dimension | 512 |
| decoder hidden dim | 1024 |
| decoder layers | 2 |
| model dropout | 0.2 |
| cell mask rate | 0.75 |
| gene mask rate | 0.25 |
| learning rate | 2e-4 |
| weight decay | 1e-8 |
| num of cluster (for GMM) | 16 |
| total parameter | 82,402,543 |

Table 5: Hyperparameters for pretraining *CellPLM* model.

| Source | Datasets |
|---|---|
| HTCA | HTAN_HTAPP, HTAN_Stanford, HTAN_Vanderbilt, HTAN_BU |
| HCA | cxg_PBMCs, EGAS00001004571_PBMCs, eQTLAutoimmune, covid19autoimmunityPBMCs, VanDerWijst-Human-10x5pv1, cxg_Airways, COMBAT2022, TabulaSapiens, PAN.A01.v01.raw_count.20210429.PFI.embedding, GTEx_8_tissues_snRNAseq_atlas_071421.public_obs |
| GEO | GSE139324, GSE136246, GSE179994,GSE131907, GSE171145, GSE139555, GSE156728_CD4, GSE148071, PMID_34663877, Qian_et_al_2020_LC, GSE176021, GSE156728_CD8 |
| Other Atlas (deduplicated) | MalteEtAl_LungAtlas, TICAtlas |

Table 6: List of dataset and data sources. External links will be included in our github repo.

## E.2 DATASETS FOR PRE-TRAINING

The dataset for pre-training contains 11.4 million cells from scRNA-seq and SRT data. scRNA-seq data consist of 4.7 million cells from human tumor cell atlas (HTCA, `https://humantumoratlas.org/`), 1.4 million cells from human cell atlas (HCA, `https://www.humancellatlas.org/`), and 2.6 million cells from Gene Expression Omnibus (GEO, `https://www.ncbi.nlm.nih.gov/geo/`). All of them are public available data, elucidated in table 6. A more detailed list and external links will be disclosed in our GitHub repository. Note that although our *CellPLM* is capable to handle various input feature sets, when we concatenated these scRNA-seq datasets, we used inner join by default of Anndata package. As a result, all scRNA-seq datasets only contain a $13,500$ common gene set. We will address this issue and increase the size of the gene set in future versions of *CellPLM*.

The SRT datasets we used are publicly available on Nanostring official website: `https://nanostring.com/products/cosmx-spatial-molecular-imager/nsclc-ffpe-dataset/`, where 2.7 million cells and $1,000$ genes are measured. Both scRNA-seq and SRT data are preprocessed with library size normalization and log1p transformation, following the convention in Stuart et al. (2019),

## F ADDITIONAL EXPERIMENTAL DETAILS

In this section, we provide more experimental details about fine-tuning, baselines, and evaluation metrics under each downstream task.

### F.1 SCRNA-SEQ DENOISING

**Downstream Task Datasets.** In scRNA-seq denoising task, we evaluate *CellPLM* on two datasets, i.e., PBMC 5K and Jurkat from 10x Genomics lin (a). It is worth noting that during the prepossessing stage, we performed sub-setting on both datasets to ensure that all the genes were included in the gene set of pre-training data. Additionally, any genes with zero counts were removed from the analysis. We list the statistics of them in Table 7.

Table 7: scRNA-seq denoising datasets

|  | 5K PBMC | Jurkat |
|---|---|---|
| Number of genes | 33,538 | 32,738 |
| Number of cells | 5,247 | 3,258 |
| Num genes picked | 7,197 | 7,618 |

**Evaluation Metrics.** Following the setting of scGNN Wang et al. (2021), scGNN2.0 Gu et al. (2022) and DeepImpute Arisdakessian et al. (2019), we performed synthetic dropout simulation with missing at random (MAR) setting. While scGNN only considered a simple scenario, i.e., randomly flipped 10% of the non-zero entries to zeros, DeepImpute applied cell-wise mask with masking probability given by a multinomial distribution. Specifically, we adapted the setting from DeepImpute with exponential kernel. For cell $i$ that contains at least 5 expressed genes, the probability that one non-zero count $x_{i,j}$ is masked during the training process is given by $\text{Exp}(0, 20)$:

$$p_{i,j} = \frac{1}{20} e^{-\frac{x}{20}},$$
$$q_{i,j} = \frac{p_{i,j}}{\sum_{j=0}^{J_i} p_{i,j}},$$

where $J_i$ is the number of non-zero counts within cell $i$. We masked 10% of the non-zero counts according to $\{q_{i,j}\}_{j=0}^{J_i}$ and evaluate model performance on the masked entries. We calculate the root mean squared error (RMSE) and mean absolute error (MAE) between the predicted values and ground truth.

**Baselines** (1) DeepImpute Arisdakessian et al. (2019) employed a strategy of dividing genes into subsets and constructing deep neural networks to impute scRNA-seq data. We implemented DeepImpute with default settings in DANCE Ding et al. (2022) package. (2) scGNN2.0 Gu et al. (2022) incorporated a feature autoencoder, a cluster autoencoder and a graph attention autoencoder for simultaneous imputation and clustering. scGNN2.0 is implemented by DANCE package with default settings. (3) GraphSCI Rao et al. (2021) combined autoencoders with graph convolution networks among a gene-gene similarity graph. We accommodated the implementation of GraphSCI in DANCE package. (4) SAVER Huang et al. (2018) leveraged Poisson LASSO regression to model the scRNA-seq counts with Poisson–gamma mixture. We utilized R package SAVER to illustrate the performance of it. (5) DCA Eraslan et al. (2019) introduced an autoencoder framework based on zero inflated negative binomial (ZINB) distribution. We applied DCA to aforementioned datasets with its Python pacakge. (6) MAGIC Van Dijk et al. (2018) utilized Markov affinity to capture gene-gene relationship and impute missing gene expression. We adapted its Python package to access the performance of it. (7) scImpute Li & Li (2018) developed a Gamma and Gaussian mixture model to identify dropout values. We revealed the performance of scImpute with its R pacakge.

**Fine-tuning.** Since denoising task requires model to recover the gene expression matrix, we can directly get the zero shot performance of *CellPLM* by specifying the gene set of target dataset. Additionally, we fine-tuned *CellPLM* by replacing the pre-trained decoder with a MLP head and initializing encoder with pre-trained weights. Additionally, for methods require model selection on validation set, we performed another 10% simulation dropout and treat masked entries as validation set. The fine-tuned *CellPLM* was trained on MSE reconstruction loss, while the best model was selected by evaluating MSE on validation set.

### F.2 SPATIAL TANSCRIPTOMIC IMPUTATION

**Downstream Task Datasets.** To evaluate spatial tanscriptomic imputation models at single-cell resolution, we collected two samples from MERSCOPE FFPE Human Immuno-oncology Data lin

(b). Specifically, we chose "Lung cancer 2" and "Liver cancer 2" as our samples, and subsequently referred to them as "Lung2" and "Liver2" respectively. The Lung2 and Liver2 datasets were subsetted to align with the gene set of the pre-training data. Additionally, we removed the fields of view (FOVs) that contained fewer than 100 cells and retained only the first 100 FOVs from both datasets. Note that all baselines require reference scRNA-seq datasets to impute the unseen genes of SRT data, we collected GSE131907 Kim et al. (2020) and GSE151530 Ma et al. (2021) for lung cancer and liver cancer, respectively. The statistics of all datasets are illustrated in Table 8.

Table 8: Spatial tanscriptomic imputation datasets.

|  | Lung2 | Liver2 | GSE131907 | GSE151530 |
|---|---|---|---|---|
| Number of genes | 500 | 500 | 29,634 | 18,667 |
| Number of cells | 836,739 | 598,141 | 208,506 | 56,721 |
| Num genes picked | 462 | 446 | All | ALL |
| Num cells picked | 40,114 | 20,629 | All | All |

**Evaluation Metrics.** Following the evaluation pipeline proposed by Avşar et al. Avşar & Pir (2023), we selected target genes of SRT data with stratified sampling according to gene sparsity. Specifically, we grouped genes into four categories: low sparse, moderate sparse, high sparse, and very-high sparse. Empirically, the boundaries were defined as $[x < 75, 75 \leq x < 90, 90 \leq x < 95, 95 \leq x]$ to approximate the Gaussian mean and standard deviation slices. Subsequently, we randomly selected 25 genes from each sparsity group and remove them from training data. After training the models, we calculate the evaluation metrics on the target genes. Namely, we compute the root mean squared error (RMSE), Pearson's correlation coefficient (PCC) and cosine similarity (Cosine) between the ground truth values and the corresponding imputed values in a gene-wise approach.

**Baselines.** (1) SpaGE Abdelaal et al. (2020) relied on domain adaptation to map scRNA-seq data onto SRT data and utilized a $k$-nearest-neighbor (k-NN) graph to predict unseen genes. We implemented SpaGE with default settings on both datasets. (2) stPlus Shengquan et al. (2021) developed an autoencoder framework for learning cell embeddings and imputing SRT genes using a weighted k-NN approach. The performance of stPlus is accessed by its Python package. (3) gimVI Lopez et al. (2019) introduced a variational autoencoder based model with protocol-specific treatments on scRNA-seq data and SRT data. We applied the scvi-tools lin (c) Python package with default settings to evaluate the performance of gimVI. (4) Tangram Biancalani et al. (2021) utilized a deep learning approach to learn the spatial alignment of scRNA-seq data based on a reference SRT dataset with consistent spatial maps. We evaluated Tangram with its Python package.

**Fine-tuning.** Similar to scRNA-seq denoising, the spatial tanscriptomic imputation task requires the ouput of the model to be the gene expression. Thus, we directly fine-tune *CellPLM* on the pre-trained weights while specifying the input genes and target genes. The last two batches were hold out for validation.

**Visualization of attention.** One essential multi-cell task is cell-cell communication (CCC) inference, where CCC mainly represents biochemical signaling through ligand-receptor binding across cells (Cang et al., 2023). Our *CellPLM* applies self-attention mechanism on cell level, from which we can study the interaction strength given by cell attention matrix. As a preliminary study, we extract the attention matrix between cells from a random chosen field of view (FOV) in Cosmx Liver dataset. The attention matrix is treated as CCC scores, and we visualize the results following the stream plot setting in Cang et al. (2023). As shown in the Figure 6 in our supplementary PDF, there are some strong trends on the left side and right side of the FOV, suggesting further exploration of specific signaling pathways for the included cells. This case study showcase the potential of our *CellPLM* model in cell-cell communication research. We hope our model can facilitate more insightful biological research in the future.

## F.3 PERTURBATION PREDICTION

The perturb-seq technology has been established to examine the gene expression response at the single-cell level when subjected to pooled perturbations (Dixit et al., 2016). By comparing the gene expression before and after perturbation, downstream analysis of differential expression (DE) enables the identification of genes that play a crucial role in disease progression. To assess the potential

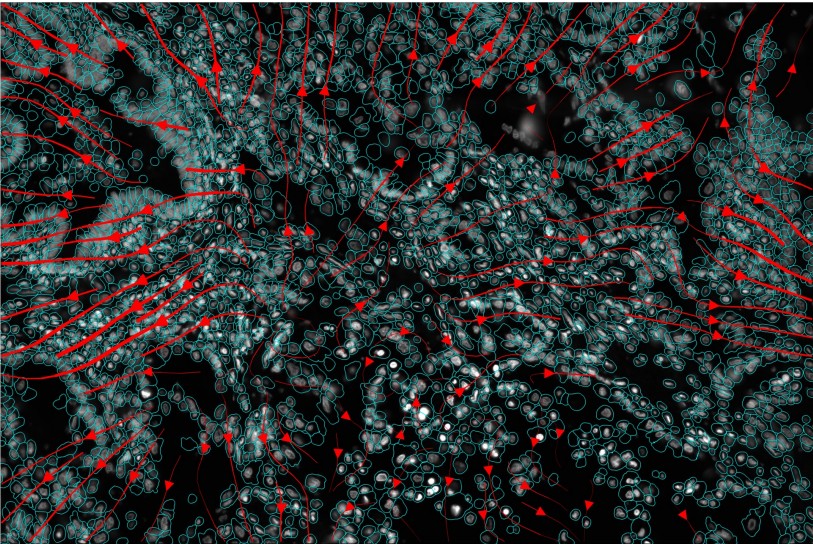

Figure 6: Visualization of attention matrix demonstrate cell-cell communication.

benefits of *CellPLM* in gene-level tasks, we conduct experiments to predict the expression value of genes after perturbation. Following the setting of GEARS (Roohani et al., 2022), we partition the perturbations into training, validation, and test sets, ensuring that none of the test perturbations are encountered during the optimization process.

Two perturbation datasets are employed for evaluation: (1) the Adamson Perturb-Seq dataset (Adamson et al., 2016), consisting of 87 one-gene perturbations; and (2) the Norman Perturb-Seq dataset (Norman et al., 2019), containing 131 two-gene perturbations and 105 one-gene perturbations. To evaluate the performance of perturbation prediction, we employ Root Mean Square Error (RMSE) to measure the degree of similarity between the non-zero ground-truth expression values and corresponding predicted gene expressions. In addition, following previous settings in GEARS (Roohani et al., 2022), we also present the RMSE calculated on the top 20 deferentially-expressed genes.

We compare the performance between *CellPLM* and two baselines, i.e., a recent preprint GEARS method (Roohani et al., 2022), and scGen (Lotfollahi et al., 2019). The results in Figure 7 imply that *CellPLM* achieves the lowest RMSE values across all settings.

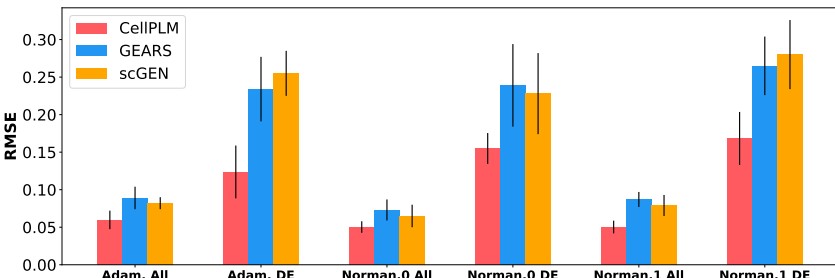

Figure 7: (*Task 3*) The RMSE performance (↓) on Adamson Perturb-Seq and the Norman Perturb-Seq datasets. The Norman Perturb-seq dataset consists of two settings: one-gene perturbations and two-gene perturbations, denoted as Norm.0 and Norm.1, respectively.

**Downstream Task Datasets.** We included the Adamson Perturb-Seq dataset Adamson et al. (2016) for one-gene perturbations and the Norman Perturb-Seq dataset Norman et al. (2019) for two-gene perturbations. We followed the preprocess pipeline of GEARS Roohani et al. (2022) and both datasets were then gene-wise subsetted to fit in the gene set of pre-training data. The statistics are summaried in Table 9.

Table 9: Perturbation prediction datasets.

|  | Adamson | Norman |
|---|---|---|
| Number of genes | 5,060 | 5,045 |
| Number of cells | 68,603 | 91,205 |
| Num genes picked | 3,246 | 2,353 |
| Num one-gene pert. | 87 | 105 |
| Num two-gene pert. | – | 131 |

**Evaluation Metrics.** Following the setting of GEARS Roohani et al. (2022), we applied data split such that the testing perturbation are unseen during the training process. Specifically, For Adamson dataset, we randomly hold out $25\%$ of the perturbations for testing and $10\%$ of the perturbations within the training set for validation. For Norman dataset, two settings for two-gene perturbations are implemented for evalutation purpose: $1/2$ unseen and $2/2$ unseen. We excluded all two-gene combinations in which at least one of the individual genes involved in the combination belonged to the unseen set. Finally, we evaluate the performance by calculating the root mean squared error (RMSE) between the predictions and the true values within the testing set.

**Baselines.** (1) GEARS Roohani et al. (2022) utilized gene co-expression knowledge graph and Gene Ontology-derived knowledge graph to model the influence of perturbations. We followed the recommended parameter settings within its Python package to access the performance. (2) scGen Lotfollahi et al. (2019) built a conditional variational autoencoders and incoporated vector arithmetics to model phenomena response. We implemented scGen with its Python package on both datasets.

**Fine-tuning.** For one perturbation, we set the input of perturbed genes to be $-100$ to mimic the gene perturbation action. During the fine-tuning process, we substituted the original batch-aware decoder with a simplified MLP decoder. Additionally, we initialized the remaining components of *CellPLM* with pre-trained weights. The final model was chosen to be the best-performed model on the validation set.

## G    CELL TYPE ANNOTATION

Cell type annotation is a crucial step in single-cell analysis as it enables the identification and characterization of distinct cell populations within a tissue or organism. This information is crucial for understanding the functional diversity, developmental trajectories, and disease relevance of different cell types, providing insights into biological processes and facilitating targeted therapeutic approaches.

**Downstream Task Datasets.** We assess the performance of *CellPLM* on the task of cell type annotation on hPancreas (Chen et al., 2023) and Multiple Sclerosis (MS) (Schirmer et al., 2019), which are suggested by Cui et al. (2023). The hPancreas dataset contains five scRNA-seq datasets of human pancreas cells, divided into reference and query sets with annotations, including 13 cell types and 11 cell types, respectively. The Multiple Sclerosis dataset (M.S.), sourced from EMBL-EBI, includes 9 healthy control and 12 M.S. samples. 3,000 highly variable genes were retained.

**Evaluation Metrics.** We evaluate cell type annotation performance based on two standard classification metrics, Macro Precision and Macro F1 score.

**Baselines.** To benchmark the performance of *CellPLM*, we compare it with both pre-trained models including scGPT Cui et al. (2023), scBERT Yang et al. (2022), as well as non-pre-trained SOTA models including ACTINN Ma & Pellegrini (2020), CellTypist Domínguez Conde et al. (2022), SingleCellNet Tan & Cahan (2019), and TOSICA Chen et al. (2023). For baseline methods, we adhere to their provided guidelines and utilize the default parameter setting. The performance metrics reported for scBERT, TOSICA and scGPT in this task are directly obtained from scGPT papers.

**Fine-tuning.** For *CellPLM* model, we attach a feed forward layer to the pre-trained encoder and latent space and tune the downstream model on the downstream dataset with a standard cross entropy loss.

# H  ADDITIONAL VISUALIZATION

## H.1  COMPARISON BETWEEN *CellPLM* AND SCVI

As a supplement to the zero-shot clustering experiments in Section 4.1, we add an additional comparison with scVI (Lopez et al., 2018) on the same dataset. As shown in Figure 8, *CellPLM* successfully outperforms scVI without any training or fine-tuning, while the latter was trained on this specific dataset.

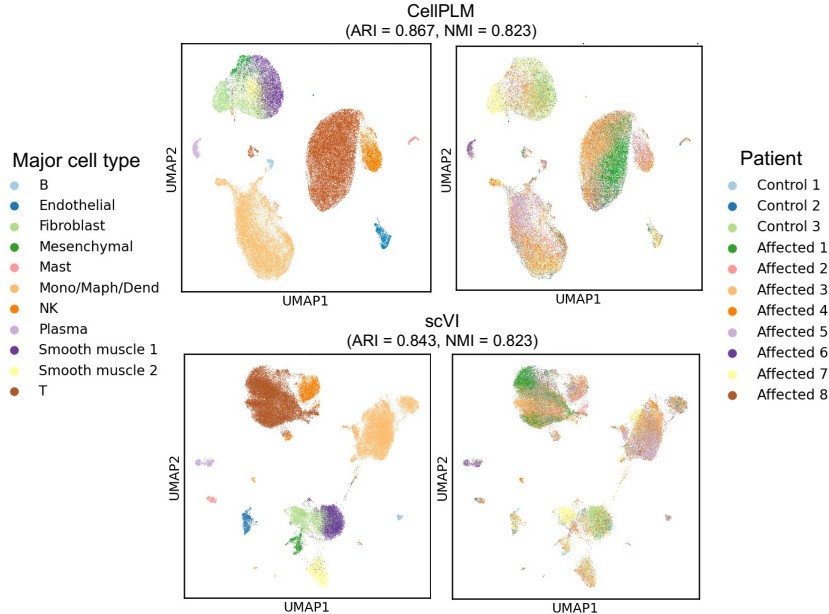

Figure 8: Visualization and comparison between *CellPLM* (zero-shot) and scVI on the clustering task.

## H.2  VISUALIZATION OF GENE EMBEDDINGS

In order to examine whether gene interactions can be encoded in *CellPLM*, we present a visualization of pre-trained gene embeddings from the gene expression embedded (i.e., $h_j$ in Eq. 3) in Figure 9. From the visualization, the gene embeddings maintain some latent structures. To further verify the effectiveness of the latent structure, we highlight a specific family of genes, HLA genes. There are multiple classes of genes in HLA gene family (Cruz-Tapias & Anaya, 2021). For example, HLA class I genes (e.g., HLA-A, -B, and -C) present endogenous peptides to responding CD8+ T Cells while the class II (e.g., HLA-DR, -DP, and –DQ) process exogenous peptides for presentation to CD4+ helper T Cells. From the UMAP visualization, HLA gene embedding clusters perfectly match the functionality and characteristics of those genes.

# I  ABLATION STUDY

To further verify the contribution of each component in *CellPLM* model, we add three new ablation studies on two representative tasks to examine the effectiveness of proposed latent distribution and transformer encoder, presented in Table 10. In each setting, we change one component in the model architecture and go through the whole pre-train and fine-tune pipeline to get the downstream performance. Specifically,

1. First, when we replace the proposed mixture of gaussian prior distribution with a gaussian prior distribution (noted as "w/o Mixture of Gaussian", commonly used in previous methods like scVI), the performance significantly drops on all datasets, indicating that an unsuitable prior distribution can greatly hurt the performance. A regular Gaussian distribution cannot accommodate the highly heterogeneous data present in the pre-train dataset, which were collected from different people, organs, and sequencing platforms.

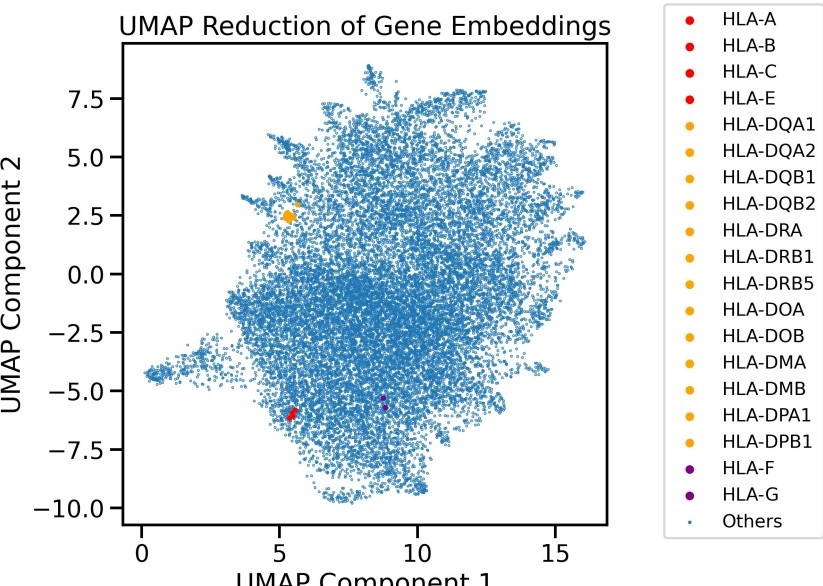

Figure 9: Visualization of gene embeddings in the pre-trained *CellPLM* demonstrate that *CellPLM* successfully captures gene interactions in the initial gene embeddings. For example, HLA Class I genes and HLA Class II perfectly form two clusters in the gene embedding space.

2. Second, we removed the latent distribution in its entirety, noted as "w/o latent distribution", i.e., we converted from a VAE-like probabilistic generative model to a deterministic masked auto-encoder. The performance consistently falls between the original one and the first ablation. On one hand, this supports our motivation of using probabilistic models with Gaussian mixture prior distribution. The latent distribution helps model the uncertainty of the data and address the high noise inherent in transcriptomic data, which results in a robust cell representation. On the other hand, the selection of prior distribution is very important because an improper prior (e.g., regular Gaussian) can be even worse than no latent distribution.

3. Lastly, we replace the transformer encoder with an MLP encoder (noted as "w/o transformer"), keeping the same number of layers and hidden dimension (the total parameters reduce from 85M to 50M). The performance significantly drops on spatial imputation task, while the gap is relatively small on cell-type classification task. This aligns with our intuition, as spatial transcriptomic data provide spatial location information, enabling the model to better identify and utilize the relationships between cells. In contrast, the cell type annotation dataset does not provide spatial location information, which makes the benefits gained from the transformer encoder more limited.

Overall, through a series of ablation studies, we have verified that our *CellPLM* model can capture the relationships between cells via the transformer encoder and enhance the performance of downstream tasks, generating more robust and useful cell representations through appropriate prior distributions.

| | Cell-type Classification | | | |
|---|---|---|---|---|
| | MS | | hPancreas | |
| | f1 | precision | f1 | precision |
| *CellPLM* | $0.766 \pm 0.007$ | $0.803 \pm 0.008$ | $0.749 \pm 0.010$ | $0.753 \pm 0.010$ |
| w/o Mixture of Gaussian | $0.737 \pm 0.042$ | $0.766 \pm 0.069$ | $0.711 \pm 0.025$ | $0.701 \pm 0.025$ |
| w/o Latent Distribution | $0.750 \pm 0.024$ | $0.809 \pm 0.032$ | $0.733 \pm 0.034$ | $0.731 \pm 0.033$ |
| w/o Transformer Encoder | $0.750 \pm 0.050$ | $0.794 \pm 0.074$ | $0.751 \pm 0.010$ | $0.750 \pm 0.012$ |
| | Spatial Imputation | | | |
| | Lung | | Liver | |
| | corr | cosine | corr | cosine |
| *CellPLM* | $0.318 \pm 0.015$ | $0.481 \pm 0.011$ | $0.328 \pm 0.011$ | $0.481 \pm 0.010$ |
| w/o Mixture of Gaussian | $0.258 \pm 0.011$ | $0.449 \pm 0.005$ | $0.232 \pm 0.013$ | $0.433 \pm 0.008$ |
| w/o Latent Distribution | $0.262 \pm 0.011$ | $0.449 \pm 0.008$ | $0.246 \pm 0.017$ | $0.428 \pm 0.012$ |
| w/o Transformer Encoder | $0.244 \pm 0.016$ | $0.443 \pm 0.008$ | $0.250 \pm 0.032$ | $0.440 \pm 0.021$ |

Table 10: Ablation studies on latent distribution and transformer encoder.

