# OpenReview forum: "CellPLM: Pre-training of Cell Language Model Beyond Single Cells"
_ICLR.cc/2024/Conference — ICLR 2024 poster_

### Official Review · Reviewer_3Haz · 2023-10-18

**Soundness:** 4 excellent
**Presentation:** 4 excellent
**Contribution:** 3 good
**Rating:** 8
**Confidence:** 4

**Summary:**

A novel pretrained single-cell transcriptomic model CellPLM is proposed. It takes cells as tokens and tissues as sentences as opposed to existing counterparts which usually takes genes as tokens and cells as sentences. By leveraging spatially-resolved transcriptomic data in pre-training, cell-cell relationships can be learned. Empirical studies demonstrate CellPLM outperforms prior arts in diverse downstream tasks and has higher efficiency in inference.

**Strengths:**

++ The study adapts the large language model pretraining techniques to the single-cell transcriptomic data, but uses a novel analogy to better capture the cell-cell relationships.

++ The paper is well organized and easy to understand. Technical details are given clearly.

++ The work provides a new direction for training single-cell foundation models and demonstrates its advantages against existing works. It will make a significant contribution to the community.

**Weaknesses:**

-- The motivation and benefit of using a VAE-like architecture is not clear.

-- Two paradigms of modeling single-cell data should both have their pros and cons but the authors do not discuss them thoroughly.

-- The benefit of using Gaussian mixture distribution as the prior of latent embedding is not validated.

-- It is not clear which parts (e.g., the combination of two types of single-cell data, the VAE-like architecture or the novel paradigm of single-cell data modeling) contribute the most to the performance gains against other existing foundation models (e.g., scGPT, scBert, etc.).

**Questions:**

1. Based on different modeling paradigms of scRNA-seq data, the proposed CellPLM may have its pros and cons compared with existing foundation models (e.g., scBert, scGPT, etc.) using genes as tokens. In which downstream tasks CellPLM may be/not be a better choice?
2. Is the batch embedding learnable? If not, how are they defined?
3. What's the motivation and benefit of using the VAE-like architecture for CellPLM when the goal is not to learn a generative model? Is there a problem if a typical transformer architecture plus a head (without Gaussian prior latent embedding) is employed?

---

> ### Author Response · Authors · 2023-11-17
> **Reply to Reviewer 3Haz**
>
> We sincerely appreciate your recognition of this work. In the following content, we will address your comments and suggestions.
>
> > W4 It is not clear which parts (e.g., the combination of two types of single-cell data, the VAE-like architecture or the novel paradigm of single-cell data modeling) contribute the most to the performance gains against other existing foundation models (e.g., scGPT, scBert, etc.).
>
> Clarifying the individual contributions of different aspects of our model to its performance is essential. To verify the contribution of each component in CellPLM model, we have now added three new ablation studies on two representative tasks to examine the effectiveness of proposed latent distribution and transformer encoder. In each setting, we change one component in the model architecture and go through the whole pre-train and fine-tune pipeline to get the downstream performance. The results are presented in Table 1 below.
>
> - First, when we replace the proposed mixture of gaussian prior distribution with a gaussian prior distribution (noted as “w/o Mixture of Gaussian”, commonly used in previous methods like scVI), the performance significantly drops on all datasets, indicating that an unsuitable prior distribution can greatly hurt the performance. A regular Gaussian distribution cannot accommodate the highly heterogeneous data present in the pre-train dataset, which were collected from different people, organs, and sequencing platforms.
> - Second, we removed the latent distribution in its entirety, noted as “w/o latent distribution”, i.e., we converted from a VAE-like probabilistic generative model to a deterministic masked auto-encoder. The performance consistently falls between the original one and the first ablation. On one hand, this supports our motivation of using probabilistic models with Gaussian mixture prior distribution. The latent distribution helps model the uncertainty of the data and address the high noise inherent in transcriptomic data. This results in a robust cell representation. On the other hand, the selection of prior distribution is very important because an improper prior (e.g., regular Gaussian) can be even worse than no latent distribution.
> - Lastly, we replace the transformer encoder with an MLP encoder, keeping the same number of layers and hidden dimension (the total parameters reduce from 85M to 50M). The performance significantly drops on spatial imputation task, while the gap is relatively small on cell-type classification task. This aligns with our intuition, as spatial transcriptomic data provide spatial location information, enabling the model to better identify and utilize the relationships between cells. In contrast, the cell type annotation dataset does not provide spatial location information, which makes the benefits gained from the transformer encoder more limited.
>
> Overall, through a series of ablation studies, we have verified that our CellPLM model can capture the relationships between cells via the transformer encoder and enhance the performance of downstream tasks, generating more robust and useful cell representations through appropriate prior distributions.
>
> ||Cell-type Classification||||Spatial Imputation||||
> |---|---|---|---|---|---|---|---|---|
> ||MS||hPancreas||Lung||Liver||
> ||Macro F1|Macro Precision|Macro F1|Macro Precision|corr|cosine|corr|cosine|
> |CellPLM|0.766$\pm$0.007|0.803$\pm$0.008|0.749$\pm$0.010|0.753$\pm$0.010|0.318$\pm$0.015|0.481$\pm$0.011|0.328$\pm$0.011|0.481$\pm$0.010|
> |w/o Mixture of Gaussian|0.737$\pm$0.042|0.766$\pm$0.069|0.711$\pm$0.025|0.701$\pm$0.025|0.258$\pm$0.011|0.449$\pm$0.005|0.232$\pm$0.013|0.433$\pm$0.008|
> |w/o Latent Distribution|0.750$\pm$0.024|0.809$\pm$0.032|0.733$\pm$0.034|0.731$\pm$0.033|0.262$\pm$0.011|0.449$\pm$0.008|0.246$\pm$0.017|0.428$\pm$0.012|
> |w/o Transformer Encoder|0.750$\pm$0.050|0.794$\pm$0.074|0.751$\pm$0.010|0.750$\pm$0.012|0.244$\pm$0.016|0.443$\pm$0.008|0.250$\pm$0.032|0.440$\pm$0.021|
> ---
> Table 1. Ablation studies of latent distribution and transformer encoder on cell-type classification and spatial imputation task.
>
> > W1. The motivation and benefit of using a VAE-like architecture is not clear.
>
> We appreciate your inquiry regarding the use of a VAE-like architecture. The choice was motivated by the need for a robust framework capable of capturing the complex, high-dimensional and noisy nature of single-cell data. The VAE architecture offers a powerful way to learn meaningful latent representations, modeling the uncertainty from the measurement, enabling the model to effectively capture underlying biological variability and noise inherent in single-cell data. As shown in the ablation studies in Table 1 above, a proper latent distribution results in more robust cell representation and better downstream performance. Due to the same reason, VAE has been a well adopted architecture in recent single-cell analysis research on various tasks[1,2,3,4,5].

---

> > ### Author Response · Authors · 2023-11-20
> > **A kindly reminder**
> >
> > Dear Reviewer 3Haz,
> >
> > We wish to express our gratitude for your thoughtful comments and concerns. Following your valuable feedback, we have submitted our responses and revisions to address the issues. As the discussion period is about to end, we kindly request your confirmation of the receipt of our responses. Additionally, we welcome any further concerns or suggestions regarding our revision. Your timely response is greatly appreciated and will be immensely helpful for us to improve our work.
> >
> > Thank you for your time and consideration.

---

> ### Author Response · Authors · 2023-11-17
> **Reply to Reviewer 3Haz #2**
>
> > W3. The benefit of using Gaussian mixture distribution as the prior of latent embedding is not validated.
>
> You concern is well-taken. As verified in the ablation studies above, the Gaussian mixture prior distribution can help model the uncertainty of the data and address the high noise inherent in transcriptomic data. It outperforms a regular Gaussian due to its capacity of handling heterogenous pre-train data. This eventually results in more robust cell representation and better downstream performance.
>
> W2. Two paradigms of modeling single-cell data should both have their pros and cons but the authors do not discuss them thoroughly.
>
> > Q1. Based on different modeling paradigms of scRNA-seq data, the proposed CellPLM may have its pros and cons compared with existing foundation models (e.g., scBert, scGPT, etc.) using genes as tokens. In which downstream tasks CellPLM may be/not be a better choice?
>
> CellPLM, with its unique approach to modeling scRNA-seq and SRT data, excels in tasks that require deep understanding of cellular heterogeneity and cell-cell relationships, such as in cell type annotation, imputation and spatial transcriptomic tasks. It also achieves impressive inference speed when generating cell embeddings. However, for tasks that explicitly requires gene-level output, CellPLM might not be capable. For example, CellPLM is not available for gene regulatory network inference, while gene-level pretrained models (e.g., scBert, scGPT, etc.) can do. Although CellPLM can implicitly capture gene interactions through gene expression embedder and encoders (as illustrated in gene embedding visualization in Appendix H.2 and gene perturbation prediction in Appendix F.3), it’s hard to query the contextualized gene representations from the model.
>
> > Q2. Is the batch embedding learnable? If not, how are they defined?
>
> Yes, it is learnable. The batch token is an important input to our decoder, which aids in batch effect removal from the latent space. The batch embeddings are randomly initialized for each tissue sample in the pre-train datasets. When transferring to new datasets in downstream tasks, we initialize the embedding of new batches with the average of all pre-train batches. The batch embeddings are then fine-tuned on the downstream datasets.
>
> Note that the cell type annotation and gene perturbation heads do not condition on batch tokens, and zero-shot clustering only relies on the encoder, thus the batch tokens are not relevant in these downstream tasks.
>
> > Q3. What's the motivation and benefit of using the VAE-like architecture for CellPLM when the goal is not to learn a generative model? Is there a problem if a typical transformer architecture plus a head (without Gaussian prior latent embedding) is employed?
>
> The use of a VAE-like architecture, despite not aiming for a generative model, brings significant advantages. It allows for a more nuanced understanding of the data's underlying distribution, particularly useful in capturing the stochastic nature of gene expression in single cells. A typical transformer architecture, while effective, may not fully capture this complexity. The inclusion of a Gaussian prior latent embedding adds an additional layer of depth in modeling the inherent variability and noise in single-cell data, which is now verified in the ablation studies in Table 1 above.
>
> ---
> [1] Lopez, Romain, et al. "Deep generative modeling for single-cell transcriptomics." *Nature methods* 15.12 (2018): 1053-1058.
>
> [2] Lopez, Romain, et al. "A joint model of unpaired data from scRNA-seq and spatial transcriptomics for imputing missing gene expression measurements." *arXiv preprint arXiv:1905.02269* (2019).
>
> [3] Grønbech, et al. "scVAE: variational auto-encoders for single-cell gene expression data." Bioinformatics (2020)
>
> [4] Xu, et al. "Graph embedding and Gaussian mixture variational autoencoder network for end-to-end analysis of single-cell RNA sequencing data." Cell Reports methods (2023).
>
> [5] Tu, Xinming, et al. "Cross-linked unified embedding for cross-modality representation learning." *Advances in Neural Information Processing Systems* 35 (2022): 15942-15955.

---

### Official Review · Reviewer_C7dG · 2023-11-01

**Soundness:** 3 good
**Presentation:** 3 good
**Contribution:** 3 good
**Rating:** 6
**Confidence:** 2

**Summary:**

The paper introduces a pre-trained language model CellPLM, the first single-cell pretrained language model that utilizes cell-cell relations.

**Strengths:**

- This paper is well written. Provide enough introductions for people without much background in this field.
- The proposed method is sound. Taking cell-cell relationships into the modeling is a well-motivated idea.

**Weaknesses:**

I integrate the questions and weaknesses in this section since I have very little bio background and all the weaknesses I identified are based on my own understanding of this area, which could be incorrect.

- Where do the huge speed improvements of CellPLM come from? It is because CellPLM aggregates the genes within each cell directly in the embedding layer, so it only needs to process dramatically shorter input sequences. So basically, CellPLM works at the cell level, while baselines work at the gene level. If this is true, the Claim "with 500x times higher inference speed compared to existing pre-trained models." is unfair since you are compared to a task that your model is specifically designed for. At least it should be reduced to "with 500x times higher inference speed when generating cell embeddings".

- Why does the CellPLM model from scratch outperform most of the pre-trained baselines? This seems to be really unrealistic to me. Is it because your model is much larger than the baselines or cell-level LM is more suitable for the tasks? If the randomly initialized model already outperforms most of the baselines, then we need to re-evaluate the claims in the paper, since the performance boost may mainly result from the Transformer architecture instead of the specific model designs you mentioned.

- There is a lack of ablation study to show the effectiveness of each proposed component.

**Questions:**

Please see the Weaknesses section.

---

> ### Author Response · Authors · 2023-11-17
> **Reply to Reviewer C7dG**
>
> We sincerely appreciate your comments on this work. In the following content, we will try to address your comments and suggestions.
>
> > W1. Where do the huge speed improvements of CellPLM come from? It is because CellPLM aggregates the genes within each cell directly in the embedding layer, so it only needs to process dramatically shorter input sequences. So basically, CellPLM works at the cell level, while baselines work at the gene level. If this is true, the Claim "with 500x times higher inference speed compared to existing pre-trained models." is unfair since you are compared to a task that your model is specifically designed for. At least it should be reduced to "with 500x times higher inference speed when generating cell embeddings".
>
> Thank you for your great suggestion. We agree that the original claim is not rigid enough. We now change it to the suggested expression in the revised paper, i.e., “with 500x times higher inference speed on generating cell embeddings compared to existing pre-trained models”. Meanwhile, we want to emphasize that the cell embeddings are essential for various downstream tasks in single-cell analysis. For example, all the downstream tasks (clustering, denoising, cell-type annotation, imputation, gene perturbation prediction) presented in the paper rely on high-quality cell representations. Therefore, the increased inference speed in generating cell embeddings should still be considered as a highlight of CellPLM model.
>
> > W2. Why does the CellPLM model from scratch outperform most of the pre-trained baselines? This seems to be really unrealistic to me. Is it because your model is much larger than the baselines or cell-level LM is more suitable for the tasks? If the randomly initialized model already outperforms most of the baselines, then we need to re-evaluate the claims in the paper, since the performance boost may mainly result from the Transformer architecture instead of the specific model designs you mentioned.
>
> We appreciate your question about the performance improvement. First, we want to clarify that CellPLM does not always outperform baselines when training from scratch. For example, in spatial imputation task (both datasets) and cell type annotation task (hPancreas dataset), it cannot outperform baselines.
>
> To further illustrate how CellPLM benefits from each component, we add an ablation study on the from-scratch version on MS dataset of cell type annotation task, presented in Table 1 below. We find that both the bi-level masking module and the transformer encoder positively contribute to the from-scratch performance. When removing these modules, CellPLM performance is closer to the non-pretrain baselines (e.g., CellTypist). Since CellPLM is the first method that applies bi-level masking and cell-level transformer encoders to these tasks, this can be considered as our contribution. Lastly, the gap between pre-trained and from-scratch is still considerably larger than the gap between from-scratch versions. This sufficiently demonstrates the benefits brought by pre-training and transfer, proving the effectiveness of the pre-training framework we proposed and laying a foundation for future downstream tasks. This is indeed our most essential contribution.
>
> |  | Macro F1 | Macro Precision |
> | --- | --- | --- |
> | CellPLM (pre-trained) | 0.766 $\pm$ 0.007 | 0.803 $\pm$ 0.008 |
> | CellPLM (from scratch) | 0.709 $\pm$ 0.007 | 0.732 $\pm$ 0.015 |
> |    w/o masks | 0.683 $\pm$ 0.013 | 0.703 $\pm$ 0.027 |
> |    w/o transformer | 0.685 $\pm$ 0.012 | 0.701 $\pm$ 0.028 |
> | CellTypist | 0.667 $\pm$ 0.002 | 0.693 $\pm$ 0.001 |
>
> Table 1. Ablation studies of mask technique and transformer encoder on MS dataset, in training from scratch scenario.

---

> > ### Author Response · Authors · 2023-11-17
> > **Reply to Reviewer C7dG #2**
> >
> > > W3. There is a lack of ablation study to show the effectiveness of each proposed component.
> >
> > We appreciate your advice. We now add three new ablation studies on two representative tasks to demonstrate the effectiveness of proposed latent distribution and transformer encoder. In each setting, we change one component in the model architecture and go through the whole pre-train and fine-tune pipeline to get the downstream performance.  The results are presented in Table 2 below.
> >
> > - First, when we replace the proposed mixture of gaussian prior distribution with a gaussian prior distribution (noted as “w/o Mixture of Gaussian”, commonly used in previous methods like scVI), the performance significantly drops on all datasets, indicating that an unsuitable prior distribution can greatly hurt the performance. A regular Gaussian distribution cannot accommodate the highly heterogeneous data present in the pre-train dataset, which were collected from different people, organs, and sequencing platforms.
> > - Second, we removed the latent distribution in its entirety, noted as “w/o latent distribution”, i.e., we converted from a VAE-like probabilistic generative model to a deterministic masked auto-encoder. The performance consistently falls between the original one and the first ablation. On one hand, this supports our motivation of using probabilistic models with Gaussian mixture prior distribution. The latent distribution helps model the uncertainty of the data and address the high noise inherent in transcriptomic data. This results in a robust cell representation. On the other hand, the selection of prior distribution is very important because an improper prior (e.g., regular Gaussian) can be even worse than no latent distribution.
> > - Lastly, we replace the transformer encoder with an MLP encoder, keeping the same number of layers and hidden dimension (the total parameters reduce from 85M to 50M). The performance significantly drops on spatial imputation task, while the gap is relatively small on cell-type classification task. This aligns with our intuition, as spatial transcriptomic data provide spatial location information, enabling the model to better identify and utilize the relationships between cells. In contrast, the cell type annotation dataset does not provide spatial location information, which makes the benefits gained from the transformer encoder more limited.
> >
> > Overall, through a series of ablation studies, we have verified that our CellPLM model can capture the relationships between cells via the transformer encoder and enhance the performance of downstream tasks, generating more robust and useful cell representations through appropriate prior distributions.
> >
> > ||Cell-type Classification||||Spatial Imputation||||
> > |---|---|---|---|---|---|---|---|---|
> > ||MS||hPancreas||Lung||Liver||
> > ||Macro F1|Macro Precision|Macro F1|Macro Precision|corr|cosine|corr|cosine|
> > |CellPLM|0.766$\pm$0.007|0.803$\pm$0.008|0.749$\pm$0.010|0.753$\pm$0.010|0.318$\pm$0.015|0.481$\pm$0.011|0.328$\pm$0.011|0.481$\pm$0.010|
> > |w/o Mixture of Gaussian|0.737$\pm$0.042|0.766$\pm$0.069|0.711$\pm$0.025|0.701$\pm$0.025|0.258$\pm$0.011|0.449$\pm$0.005|0.232$\pm$0.013|0.433$\pm$0.008|
> > |w/o Latent Distribution|0.750$\pm$0.024|0.809$\pm$0.032|0.733$\pm$0.034|0.731$\pm$0.033|0.262$\pm$0.011|0.449$\pm$0.008|0.246$\pm$0.017|0.428$\pm$0.012|
> > |w/o Transformer Encoder|0.750$\pm$0.050|0.794$\pm$0.074|0.751$\pm$0.010|0.750$\pm$0.012|0.244$\pm$0.016|0.443$\pm$0.008|0.250$\pm$0.032|0.440$\pm$0.021|
> > ---
> > Table 2. Ablation studies of latent distribution and transformer encoder on cell-type classification and spatial imputation task.

---

> ### Author Response · Authors · 2023-11-20
> **A kindly reminder**
>
> Dear Reviewer C7dG,
>
> Thank you for taking the time to review our work. We appreciate your feedback and we have prepared a thorough response to address your concerns. We believe that we have responded to and addressed all your concerns with our revisions — in light of this, we hope you consider raising your score. Please let us know in case there are outstanding concerns, and if so, we will be happy to respond.
>
> Notably, given that we are approaching the deadline for the rebuttal phase, we hope we can have the discussion soon. Thanks!

---

> > ### Comment · Reviewer_C7dG · 2023-11-20
> > **I have raised my score to 6**
> >
> > Thanks for your response. It solves my concerns and I have raised my score to 6.

---

> > > ### Author Response · Authors · 2023-11-20
> > > **Thanks for your response**
> > >
> > > Thank you for your response and support. We're pleased to hear that our rebuttal effectively addressed your concerns. Should any further issues arise, please do not hesitate to inform us.
> > >
> > > Additionally, we noticed that your **rating** in the review might not have been updated yet. We would greatly appreciate it if you could take a moment to revise it. Thank you!

---

> > > > ### Comment · Reviewer_C7dG · 2023-11-21
> > > > **Score changed**
> > > >
> > > > Thanks for all your responses and for this remind. Now the rating is changed.

---

### Official Review · Reviewer_1rfQ · 2023-11-02

**Soundness:** 3 good
**Presentation:** 3 good
**Contribution:** 3 good
**Rating:** 6
**Confidence:** 2

**Summary:**

In this paper, the authors propose a CellPLM to address certain issues identified in previous work. Specifically, the proposed CellPLM considers cells as tokens and tissues as sentences, whereas previous work treated genes as tokens and cells as sentences. The CellPLM is designed to learn the relationship between cells. Additionally, they adopt a Gaussian mixture prior distribution to overcome the out-of-distribution problem. Experimental results show CellPLM consistently  outperforms previous work.

**Strengths:**

1. The proposed method effectively addresses the issues contained in previous work.

2. Experimental results demonstrate that CellPLM consistently surpasses the performance of previous methods.

**Weaknesses:**

1. The experiments in this paper are not sufficient. For instance, ablation studies are needed to verify the effectiveness of each module.

2. Although the improvement in experimental results is very significant, further analysis is needed to determine whether it is due to the relationships between cells.

3. I'm not sure if the Gaussian distribution can achieve the expected effect in this method, this point also needs to be validated through experiments.

**Questions:**

see Weaknesses

---

> ### Author Response · Authors · 2023-11-17
> **Reply to Reviewer 1rfQ**
>
> We sincerely appreciate your recognition of this work. In the following content, we will try to address your comments and suggestions.
>
> > W1. The experiments in this paper are not sufficient. For instance, ablation studies are needed to verify the effectiveness of each module.
>
> We highly appreciate your advice. We now add three new ablation studies on two representative tasks to demonstrate the effectiveness of proposed latent distribution and transformer encoder. In each setting, we change one component in the model architecture and go through the whole pre-train and fine-tune pipeline to get the downstream performance.  The results are presented in Table 1 below.
>
> - First, when we replace the proposed mixture of gaussian prior distribution with a gaussian prior distribution (noted as “w/o Mixture of Gaussian”, commonly used in previous methods like scVI), the performance significantly drops on all datasets, indicating that an unsuitable prior distribution can greatly hurt the performance. A regular Gaussian distribution cannot accommodate the highly heterogeneous data present in the pre-train dataset, which were collected from different people, organs, and sequencing platforms.
> - Second, we removed the latent distribution in its entirety, noted as “w/o latent distribution”, i.e., we converted from a VAE-like probabilistic generative model to a deterministic masked auto-encoder. The performance consistently falls between the original one and the first ablation. On one hand, this supports our motivation of using probabilistic models with Gaussian mixture prior distribution. The latent distribution helps model the uncertainty of the data and address the high noise inherent in transcriptomic data, which results in a robust cell representation. On the other hand, the selection of prior distribution is very important because an improper prior (e.g., regular Gaussian) can be even worse than no latent distribution.
> - Lastly, we replace the transformer encoder with an MLP encoder, keeping the same number of layers and hidden dimension (the total parameters reduce from 85M to 50M). The performance significantly drops on spatial imputation task, while the gap is relatively small on cell-type classification task. This aligns with our intuition, as spatial transcriptomic data provide spatial location information, enabling the model to better identify and utilize the relationships between cells. In contrast, the cell type annotation dataset does not provide spatial location information, which makes the benefits gained from the transformer encoder more limited.
>
> Overall, through a series of ablation studies, we have verified that our CellPLM model can capture the relationships between cells via the transformer encoder and enhance the performance of downstream tasks, generating more robust and useful cell representations through appropriate prior distributions.
>
> ||Cell-type Classification||||Spatial Imputation||||
> |---|---|---|---|---|---|---|---|---|
> ||MS||hPancreas||Lung||Liver||
> ||Macro F1|Macro Precision|Macro F1|Macro Precision|corr|cosine|corr|cosine|
> |CellPLM|0.766$\pm$0.007|0.803$\pm$0.008|0.749$\pm$0.010|0.753$\pm$0.010|0.318$\pm$0.015|0.481$\pm$0.011|0.328$\pm$0.011|0.481$\pm$0.010|
> |w/o Mixture of Gaussian|0.737$\pm$0.042|0.766$\pm$0.069|0.711$\pm$0.025|0.701$\pm$0.025|0.258$\pm$0.011|0.449$\pm$0.005|0.232$\pm$0.013|0.433$\pm$0.008|
> |w/o Latent Distribution|0.750$\pm$0.024|0.809$\pm$0.032|0.733$\pm$0.034|0.731$\pm$0.033|0.262$\pm$0.011|0.449$\pm$0.008|0.246$\pm$0.017|0.428$\pm$0.012|
> |w/o Transformer Encoder|0.750$\pm$0.050|0.794$\pm$0.074|0.751$\pm$0.010|0.750$\pm$0.012|0.244$\pm$0.016|0.443$\pm$0.008|0.250$\pm$0.032|0.440$\pm$0.021|
> ---
> Table 1. Ablation studies of latent distribution and transformer encoder on cell-type classification and spatial imputation task.

---

> > ### Author Response · Authors · 2023-11-17
> > **Reply to Reviewer 1rfQ #2**
> >
> > > W2. Although the improvement in experimental results is very significant, further analysis is needed to determine whether it is due to the relationships between cells.
> >
> > Thank you for the suggestion. As verified in the ablation studies in Table 1 above, the model does leverage the cell-cell relations via the transformer encoder. With the help of transformer encoder, on cell classification tasks, the F1 score increases from 0.75 to 0.766 on MS dataset, and on spatial imputation tasks, the correlation increases from 0.244 to 0.318 on Lung dataset.
> >
> > > W3. I'm not sure if the Gaussian distribution can achieve the expected effect in this method, this point also needs to be validated through experiments.
> >
> > Thank you for your suggestion. As verified in the ablation studies in Table 1 above, the Gaussian mixture prior distribution is more suitable than a regular Gaussian prior distribution, which can help model the uncertainty of the data and address the high noise and heterogeneity inherent in transcriptomic data. This eventually results in more robust cell representation and better downstream performance. In the experiments, when pre-training with Gaussian mixture prior distribution, the F1 score increases from 0.75 to 0.766 on MS dataset on cell classification task, and the correlation increases from 0.262 to 0.318 on Lung dataset on spatial imputation tasks.

---

> ### Author Response · Authors · 2023-11-20
> **A kindly reminder**
>
> Dear Reviewer 1rfQ,
>
> Thank you for taking the time to review our work. We appreciate your feedback and we have prepared a thorough response to address your concerns. We believe that we have responded to and addressed all your concerns with our revisions — in light of this, we hope you consider raising your score. Please let us know in case there are outstanding concerns, and if so, we will be happy to respond.
>
> Notably, given that we are approaching the deadline for the rebuttal phase, we hope we can have the discussion soon. Thanks!

---

### Official Review · Reviewer_QJ4u · 2023-11-12

**Soundness:** 3 good
**Presentation:** 3 good
**Contribution:** 3 good
**Rating:** 6
**Confidence:** 4

**Summary:**

The authors propose a language model that learns from single-cell RNA sequencing (scRNA-seq) data. Their model takes as input a series of genes and their expression values, aggregates them, and then feeds them through an embedding matrix to create cell-level embeddings. These embeddings are then fed to a transformer along with other cell embeddings. The model is trained using the variational autoencoder (VAE) objective with a Gaussian prior, which includes a reconstruction term, a conditional prior term, and a cluster prior term. Unlike previous models, the authors leverage spatial data and show that their model outperforms previous models on several tasks.

Further, they claim that their method is faster than previous methods -- the reason for this is because of collapsing the model input down to cell level representations as opposed to feeding individual gene level tokens

**Strengths:**

Overall, the paper is clear and easy to follow, but several important details must be included in the text, such as the prior estimation for $z_i$ and $y_i$, and the dataset details and experimental setup. It is also important to note how the comparisons were standardized.

Other strengths of the paper include:

* It explores spatial + gene expression data using LLMs.
* It models the task at a cell level as opposed to a gene level (questions in weakness).
* It explores a smooth latent space for this task.

**Weaknesses:**

The source of the pre-training gains is unclear, given the current presentation. The authors restrict the gene set to a subset of 13,500 genes, but it is not specified which genes are included or whether they are protein-coding or non-protein-coding. The dataset section should be expanded to include details on how the data was preprocessed.

* **Cell masking:** The authors aggregate cell genes using their embeddings and pass these tokens to the LLM. What does it mean to mask a cell? In the diagram, the cells are masked, but in the equations, the genes are masked. Assuming the masking is at the gene level and samples are taken from the same batch, wouldn't other cell genes have the necessary information to impute the cell? In essence, is the model cheating?
* **Single-cell operation:** How would the model operate if you only have a single-cell sample? Would you feed a single cell token to the LLM and use the latent representation?
* **Sequence length:** I'm curious how the model performance would change if you vary the "sequence length". For example, in your batch, do you have the same number of genes for each cell?
* **Spatial information:** How is the spatial information leveraged here? Is it through the positional encoding? What would happen if this positional encoding is removed?
* **Positional Encoding:** It is not clear if this had a significant improvement in the quality of representation learnt.
* **Comparison with previous models:** Although the authors compare with previous single-cell models, these models only operated on gene expression data and fed a single cell at a time (i.e., the sequence dimension is the genes within the cell as opposed to other cells, as done in this work).
* **Benefit of pre-training:** The benefit of which part of the pre-training is the result of the improvements shown in the result section is not clear from the current presentation. Also, the authors don't compare with scVI (although they do cite the method) or even a simpler baseline like HVG.
* **Gene interactions:** Does the model learn gene level interactions? By feeding cell-level representations, how can this be assessed?
* **Batch Token:** How would you transfer the model to a dataset which has a new batch token are these fine-tuned as well?

**Questions:**

See weakness section

---

> ### Author Response · Authors · 2023-11-17
> **Reply to Reviewer QJ4u**
>
> We sincerely appreciate your recognition of this work. In the following content, we will address your comments and suggestions.
>
> > W1 The source of the pre-training gains is unclear, given the current presentation. The authors restrict the gene set to a subset of 13,500 genes, but it is not specified which genes are included or whether they are protein-coding or non-protein-coding. The dataset section should be expanded to include details on how the data was preprocessed.
>
> Thank you for highlighting the need for clarity in our data processing. We appreciate the opportunity to provide more details. In our study, the subset of 13,500 genes was resulted from the intersection of all scRNA-seq pre-train datasets, based on their prevalence and relevance in the existing literature. Most of these genes are protein-coding genes. A more detailed list of datasets is now included in Appendix E.2.  All data are pre-filtered (i.e., quality control) by the original authors and are collected in AnnData objects. We further perform normalization and log transformation following the convention in Seurat[1]. This important detail is now added to Section 3.2 and Appendix E.2.
>
> > W2 Cell masking: The authors aggregate cell genes using their embeddings and pass these tokens to the LLM. What does it mean to mask a cell? In the diagram, the cells are masked, but in the equations, the genes are masked. Assuming the masking is at the gene level and samples are taken from the same batch, wouldn't other cell genes have the necessary information to impute the cell? In essence, is the model cheating?
>
> Your question about cell masking is insightful. In our approach, cell masking refers to the process of masking at the gene level within each cell, same as the “bi-level masking technique” proposed in [2]. The core idea is to mask most genes (e.g., 75%) in a subset of cells (e.g., 25%), forcing the model to leverage information from other cells to impute the missing information. Therefore, your understanding of “other cell genes have the necessary information to impute the cell” is totally correct.
>
> **Our method is not deemed “cheating” as it uses the same amount of input as other models and does not use any label information from the test set.**  In fact, leveraging intercellular information is an essential idea of our proposed “cell language modeling”. In addition to utilizing intracellular partially observed gene expression data, our model is also trained to identify and exploit similar cells within the same tissue, cellular communication, and spatial neighborhood composition to impute indeed information. This capability is essential for better denoising and spatial-resolved transcriptomic applications, and previous single-cell pre-trained models are not capable of doing this.
>
> > W3 Single-cell operation: How would the model operate if you only have a single-cell sample? Would you feed a single cell token to the LLM and use the latent representation?
>
> For a single cell, the model indeed operates by feeding a single cell token into the model, utilizing its latent representation (the performance is shown in Table 1 below, in our response to “W4 Sequencing Length”). This process is similar to handling larger samples but adapted to maximize information extraction from a single-cell context. However, our model is more suitable for batch processing of data, which is in line with the characteristics of high-throughput sequencing data.
>
> > W4 Sequence length: I'm curious how the model performance would change if you vary the "sequence length". For example, in your batch, do you have the same number of genes for each cell?
>
> The question of varying sequence length is pertinent. In our framework, the number of genes per cell can be variable.  This is managed by the gene expression embedder, which is implemented by a sparse aggregation operation, and thus is flexible to any input gene sets (within the pretrain gene set).
>
> However, we want to emphasize that the “sequence length” in our transformer model does not refer to the number of genes but the number of cells. We treat cells as tokens and use the transformer model to process a batch of cells. Therefore, the “sequence length” in CellPLM is equivalent to “batch size” in traditional deep learning models. To evaluate the impact of “sequence length” (a.k.a, “batch size”), we separate the input dataset (MS dataset for cell-type annotation task) into smaller batches and run a same fine-tuned cell-type annotation model for inference. The result is presented in the table 1 below. From the experiments, the performance of the model is not sensitive to batch size, but too small a batch size can lead to slight performance degradation.
>
> |Batch Size|32768|2048|128|32|16|8|4|2|1|
> |---|---|---|---|---|---|---|---|---|---|
> |Macro F1| 0.773 | 0.773 | 0.773 | 0.773 | 0.770 | 0.768 | 0.769 | 0.767 | 0.768 |
> ---
> Table 1. Parameter analysis of batch size on MS dataset for cell-type annotation task.

---

> > ### Author Response · Authors · 2023-11-17
> > **Reply to Reviewer QJ4u #2**
> >
> > > W5 Spatial information: How is the spatial information leveraged here? Is it through the positional encoding? What would happen if this positional encoding is removed?
> >
> > > W6 Positional Encoding: It is not clear if this had a significant improvement in the quality of representation learnt.
> >
> > Spatial information in our model is indeed leveraged through positional encoding, with the help of transformer encoders. We appreciate the need to clarify its impact. To address this, we conducted additional experiments to fine-tune the same pre-trained model with and without positional encoding on spatial imputation task, the results of which are presented in the Table 2 below.
> >
> > ||Lung||Liver||
> > |---|---|---|---|---|
> > ||corr|cosine|corr|cosine|
> > |CellPLM|0.318$\pm$0.015|0.481$\pm$0.011|0.328$\pm$0.011|0.481$\pm$0.010|
> > |w/o Positional Encoding|0.237$\pm$0.027|0.442$\pm$0.007|0.215$\pm$0.029|0.424$\pm$0.016|
> > ---
> > Table 2. Ablation study of positional encoding on spatial imputation task.
> >
> > This ablation experiment demonstrates the significance of positional encoding in enhancing the performance on spatial transcriptomic data, indicates that the model can leverage the spatial information from positional encoding.
> >
> > > W7 Comparison with previous models: Although the authors compare with previous single-cell models, these models only operated on gene expression data and fed a single cell at a time (i.e., the sequence dimension is the genes within the cell as opposed to other cells, as done in this work).
> >
> > We thank you for pointing out the distinction between CellPLM and existing models. As we introduced in Section 2, the existing pre-trained models all focus solely on modeling gene relationships within individual cells, neglecting the intercellular information in an organism. CellPLM is the first pre-trained model that can leverage this important information inherent in the high-throughput sequencing data and spatially-resolved data. We highlighted this difference in Figure 1 in the paper.
> >
> > In addition, cell-cell relations have been leveraged in non-pretrained baselines. For example, graph-based models (e.g., GraphSCI and scGNN, stPlus, SpaGE) in denoising and imputation tasks explicitly construct **cell-cell nearest-neighbor graphs** to leverage such information. Compared to those methods, CellPLM adopts a transformer to implicitly model the cell-cell relations.

---

> ### Author Response · Authors · 2023-11-17
> **Reply to Reviewer QJ4u #3**
>
> > W8 Benefit of pre-training: The benefit of which part of the pre-training is the result of the improvements shown in the result section is not clear from the current presentation. Also, the authors don't compare with scVI (although they do cite the method) or even a simpler baseline like HVG.
>
> Clarifying the specific benefits of pre-training is crucial, and we thank you for emphasizing this. The direct evidence of the benefit of **knowledge transfer** from pre-training is the gap between from-scratch performance and fine-tune performance as we presented in Table 2, 3 and 4. On all tasks and all datasets, pre-train fine-tune outperforms training CellPLM models directly on downstream tasks (a.k.a, “from scratch”).
>
> To further verify the contribution of each **architectural component** in CellPLM model, we have now added three new ablation studies on two representative tasks to examine the effectiveness of proposed latent distribution and transformer encoder. In each setting, we change one component in the model architecture and go through the whole pre-train and fine-tune pipeline to get the downstream performance. The results are presented in Table 3 below and added into Appendix I.
>
> - First, when we replace the proposed mixture of gaussian prior distribution with a gaussian prior distribution (noted as “w/o Mixture of Gaussian”, commonly used in previous methods like scVI), the performance significantly drops on all datasets, indicating that an unsuitable prior distribution can greatly hurt the performance. A regular Gaussian distribution cannot accommodate the highly heterogeneous data present in the pre-train dataset, which were collected from different people, organs, and sequencing platforms.
> - Second, we removed the latent distribution in its entirety, noted as “w/o latent distribution”, i.e., we converted from a VAE-like probabilistic generative model to a deterministic masked auto-encoder. The performance consistently falls between the original one and the first ablation. On one hand, this supports our motivation of using probabilistic models with Gaussian mixture prior distribution. The latent distribution helps model the uncertainty of the data and address the high noise inherent in transcriptomic data, which results in a robust cell representation. On the other hand, the selection of prior distribution is very important because an improper prior (e.g., regular Gaussian) can be even worse than no latent distribution.
> - Lastly, we replace the transformer encoder with an MLP encoder, keeping the same number of layers and hidden dimension (the total parameters reduce from 85M to 50M). The performance significantly drops on spatial imputation task, while the gap is relatively small on cell-type classification task. This aligns with our intuition, as spatial transcriptomic data provide spatial location information, enabling the model to better identify and utilize the relationships between cells. In contrast, the cell type annotation dataset does not provide spatial location information, which makes the benefits gained from the transformer encoder more limited.
>
> ||Cell-type Classification||||Spatial Imputation||||
> |---|---|---|---|---|---|---|---|---|
> ||MS||hPancreas||Lung||Liver||
> ||Macro F1|Macro Precision|Macro F1|Macro Precision|corr|cosine|corr|cosine|
> |CellPLM|0.766$\pm$0.007|0.803$\pm$0.008|0.749$\pm$0.010|0.753$\pm$0.010|0.318$\pm$0.015|0.481$\pm$0.011|0.328$\pm$0.011|0.481$\pm$0.010|
> |w/o Mixture of Gaussian|0.737$\pm$0.042|0.766$\pm$0.069|0.711$\pm$0.025|0.701$\pm$0.025|0.258$\pm$0.011|0.449$\pm$0.005|0.232$\pm$0.013|0.433$\pm$0.008|
> |w/o Latent Distribution|0.750$\pm$0.024|0.809$\pm$0.032|0.733$\pm$0.034|0.731$\pm$0.033|0.262$\pm$0.011|0.449$\pm$0.008|0.246$\pm$0.017|0.428$\pm$0.012|
> |w/o Transformer Encoder|0.750$\pm$0.050|0.794$\pm$0.074|0.751$\pm$0.010|0.750$\pm$0.012|0.244$\pm$0.016|0.443$\pm$0.008|0.250$\pm$0.032|0.440$\pm$0.021|
> ---
> Table 3. Ablation studies of latent distribution and transformer encoder on cell-type classification and spatial imputation task.
>
> In addition, the result of **scVI** is now added in the denoising task (Table 2 in Section 4.2) using `get_normalized_expression` function of scVI package and the performance of scVI is very close to DCA. Meanwhile, we want to clarify that the PCA baseline in zero-shot clustering experiments (in Section 4.1) indeed come from **HVG**. As we mentioned in Section 4.1, the PCA results are obtained from 4,500 highly variable genes (HVG). We also include a new clustering result with **scVI** on the same dataset. It achieves ARI=0.843, NMI=0.823, slightly outperforming HVG+PCA baseline but still worse than CellPLM (zero-shot). The visualization is added in the Appendix H.1.

---

> ### Author Response · Authors · 2023-11-17
> **Reply to Reviewer QJ4u #4**
>
> > W9 Gene interactions: Does the model learn gene level interactions? By feeding cell-level representations, how can this be assessed?
>
> Thank you for the great question about gene-level interactions. Our model, despite feeding cell-level representations, is designed to capture gene-level interactions through its inherent architecture. The gene functionality is first captured by gene embeddings in the expression embedder module, and then interacted through the feed-forward networks in the transformer encoder, coupled with cell-cell interaction.
>
> To evaluate whether the model learns meaningful gene representations, we provide a new visualization of gene embeddings in appendix H.2. From the visualization, the gene embeddings maintain some latent structures. As a case study, we highlight a specific family of genes, HLA genes. There are multiple classes of HLA genes[3]. For example, HLA class I genes (e.g., HLA-A, -B, and -C) present endogenous peptides to responding CD8+ T Cells while the class II (e.g., HLA-DR, -DP, and –DQ) process exogenous peptides for presentation to CD4+ helper T Cells. From the UMAP visualization, HLA gene embeddings present clusters that perfectly match the functionality and characteristics of those genes. To further assess whether the model implicitly encode gene-gene interactions, we conduct experiments of gene perturbation prediction (presented in appendix F.3), which demonstrates that our model is able to identify the knock-out effect of specific genes.
>
> > W10 Batch Token: How would you transfer the model to a dataset which has a new batch token are these fine-tuned as well?
>
> The batch token is an important input to our decoder. Currently, when transferring to new datasets, we initialize the embedding of new batches with the average of all pre-train batches. The batch embeddings are then fine-tuned during the transfer process.
>
> Note that the cell type annotation and gene perturbation heads do not condition on batch tokens, and zero-shot clustering only relies on the encoder, thus the batch tokens are not relevant in these tasks.
>
> ---
> [1] Stuart, Tim, et al. "Comprehensive integration of single-cell data." *Cell* 177.7 (2019): 1888-1902.
>
> [2] Wen, Hongzhi, et al. "Single Cells Are Spatial Tokens: Transformers for Spatial Transcriptomic Data Imputation." *arXiv preprint arXiv:2302.03038* (2023).
>
> [3] Cruz-Tapias, P. C. J., and J. M. Anaya. "Major histocompatibility complex: Antigen processing and presentation 2013." (2021).

---

> ### Author Response · Authors · 2023-11-20
> **A kindly reminder**
>
> Dear Reviewer QJ4u,
>
> Thank you for taking the time to review our work. We appreciate your feedback and we have prepared a thorough response to address your concerns. We believe that we have responded to and addressed all your concerns with our revisions — in light of this, we hope you consider raising your score. Please let us know in case there are outstanding concerns, and if so, we will be happy to respond.
>
> Notably, given that we are approaching the deadline for the rebuttal phase, we hope we can have the discussion soon. Thanks!

---

> ### Author Response · Authors · 2023-11-22
> **A kindly reminder**
>
> We appreciate your comments. We hope that our responses have adequately addressed your concerns. As the open discussion period will **conclude in one day**, we kindly remind you to share any additional concerns you may have. We welcome further discussion and are eager to engage in constructive dialogue.

---

> ### Author Response · Authors · 2023-11-23
> **A kindly reminder**
>
> Dear Reviewer QJ4u,
>
> the discussion period will end in 2 hours. If you have any further concerns, please do not hesitate to take advantage of the last moment to tell us. Thank you!

---

### Meta-Review · Program_Chairs · 2024-01-15

**Metareview:**

This paper proposes a novel model for single-cell transformer pretraining. It identifies the issues of directly using existing NLP techniques, which treats genes as tokens and cells as sentences. Instead, the proposed approach CellPLM treats cells as tokens and tissues as sentences, and then further capture the spatial relation between cells. It also addresses the issue where single-cell data are very noisy via a VAE objective. The results have demonstrated a superior performance compared to baselines. The authors are very responsive in the rebuttal stage, clarifying technique details such as mask design and addressing concerns of reviewers about effectiveness of each module by adding more ablation studies. Two reviewers raised their scores after the rebuttal.

**Justification For Why Not Higher Score:**

It's a very successful execution of a neat idea. In terms of contribution, it is based on existing framework.

**Justification For Why Not Lower Score:**

The reviewers are in general supporting this paper and happy with the rebuttal.

---

### Decision · Program_Chairs · 2024-01-16

Accept (poster)